



# Autocalibration of a physically-based hydrological model: does it produce physically realistic parameters?

Eleyna L. McGrady[1], Stephen J. Birkinshaw[1], Elizabeth Lewis[2], Ben A. Smith[1], Claire L. Walsh[1], Geoff Darch[3], Jeremy Dearlove[4]

[1]School of Engineering, Newcastle University, Newcastle-upon-Tyne, NE1 7RU, United Kingdom
[2]School of Engineering, The University of Manchester, Manchester, M13 9PL, United Kingdom
[3]Anglian Water Services, Thorpe Wood House, Thorpe Wood, Peterborough, PE3 6WT, United Kingdom
[4]Northumbrian Water Ltd, Northumbria House, Abbey Rd, Pity Me, Durham, DH1 5FJ, United Kingdom

*Correspondence to*: Eleyna L. McGrady (e.mcgrady@newcastle.ac.uk)

**Abstract.** Hydrological models are essential tools when predicting water availability, floods, and droughts. Physically-based models are capable of representing sophisticated degrees of realism compared to conceptual or data-driven models as they explicitly solve equations based on well-established physical laws that are directly related to catchment processes. However, they can require extensive calibration, which can be computationally demanding. This study develops and applies an autocalibration method for SHETRAN, a physically-based model, to improve its performance across 698 catchments in the UK. This paper discusses the process of model calibration, the benefits and caveats of the approach and discuss the extent to which physical realism of the parameters are preserved through the autocalibration.

Results show that the autocalibration process significantly improves SHETRAN's performance, raising the median NSE value for the 698 catchments from 0.69 to 0.82. After calibration, 85% of catchments achieve NSE values of ≥0.7, demonstrating a substantial enhancement in accuracy of simulations across a range of catchments with different climatic, hydrological, topographical, and geological characteristics. The greatest improvements were observed in groundwater-dominated catchments, where uncalibrated simulations struggled. Additionally, simulated transmissivity values align well with measured data, providing confidence in the model's ability to produce parameters that mirror physical realism.

This study highlights the feasibility of applying physically-based models at a national scale when combined with effective autocalibration techniques. Autocalibrated-SHETRAN-UK performs comparably to conceptual and data-driven models, whilst offering improved transparency of hydrological processes. Future work will focus on integrating groundwater levels into the calibration process of SHETRAN and refining the model by introducing more spatial complexity in soil and aquifer representation within the model to better reflect real-world variability. These advancements will further enhance our capability to simulate hydrological responses under changing climatic and land-use conditions using SHETRAN.



# 1 Introduction

Hydrological models are important tools that are used by water managers globally to provide insights into water in the environment and hydrological risks – namely, floods and droughts. They play a part in forecasting and ultimately an important role informing water management decisions and policies.

The development of hydrological models at a catchment scale is challenging due to the complexity of hydrological processes, with spatial variations in the characteristics of soil, geology and land use, in addition to temporal and spatial variability of climatic conditions (Beven, 2012). However, hydrological models act as a tool to facilitate understanding of how physical changes e.g. land-use change, and climatic changes could affect catchment hydrological processes and water

availability caused by either natural or anthropogenic changes (Özdoğan-Sarıkoç and Dadaser-Celik, 2024). Reviews of hydrological models (Keller et al., 2023; Paul et al., 2021; Peel and McMahon, 2020) show they can largely be categorised into three different groups: data-driven, conceptual, and physically based. The use of data driven models has increased in popularity in recent decades with improvements in computational processing. They can be trained easily without knowledge of the physical processes occurring within the catchment, and instead are calibrated using long-term dependencies between

meteorological data and outlet discharge (Huntingford et al., 2019; Lees et al., 2021). Conceptual models simulate hydrological processes within catchments through linked storage units which have fluxes in and out, defined by empirical relationships. These types of models are commonly used and have received significant development over the past half century (Peel and McMahon, 2020). Both data-driven and conceptual models are generally simple to understand and quick to run, whilst still producing accurate simulations. Consequently, they have often been applied in national scale hydrological

studies (Coxon et al., 2019; Lane et al., 2019; Lees et al., 2021). However, these models rely heavily on observed data to achieve a good fit between measured and simulated discharges and hence often produce parameters that have no physical basis when compared to measured values within catchments.

Physically-based models, such as SHETRAN (Ewen et al., 2000), MIKE SHE (Ma et al., 2016), ParFlow (Kollet and

Maxwell, 2008), and HydroGeoSphere (Brunner and Simmons, 2012) are capable of representing higher degrees of realism by explicitly solving equations based on well-established physical laws directly related to catchment processes (Freeze and Harlan, 1969). Consequently, these models offer several advantages over less complex models (Fatichi et al., 2016). For instance, the parameters in physically-based models are derived from fundamental physical processes and therefore enable a more accurate representation of hydrological processes. Additionally, these models can consider the spatial variations in land

use, soil properties, and input variables such as rainfall and evapotranspiration, therefore allowing, for example, the effects of changes in land use on the hydrological cycle to be considered (Islam, 2011). Moreover, they can be used to extrapolate results for scenarios that are outside of the range of conditions considered for calibration (Herrera et al., 2022). Despite the advantages that physically-based models have in modelling the environment, fully-integrated, physically-based models have



generally not been applied to large-scale national studies due to extensive data and computational resource requirements.
Calibrating and validating fully-integrated, physically-based models requires a considerable amount of time and the optimum
parameter space is rarely achieved due to user error and uncertainty (Beven, 2001).

Physically-based model calibration and parameter estimation is usually performed using a "trial and error" method where
parameters are adjusted manually until simulated and observed river runoffs match well. Whilst this approach often works
well, automatic parameter estimation tools can significantly aid the calibration process in both accuracy and time. Numerous
different algorithms have been investigated to allow for optimisation and automation of parameter estimation in hydrological
models (Kunstmann et al., 2006). These include, but are not limited to, the Simulated Annealing Method (Kirkpatrick et al.,
1983), the Shuffled Complex Evolution Global Optimization Algorithm (SCEUA) (Duan et al., 1992, 1994, 1993), the
shuffled Complex Evolution Metropolis method (SCEM-UA) (Vrugt et al., 2003), and the multi-objective complex evolution
algorithm (MOCOM) (Yapo et al., 1998). The requirement of estimating the parameters of physically-based models has long
been recognised as a potential limitation of this type of model (Herrera et al., 2022), and has been a reason to consider them
as sophisticated lumped models (Beven, 1989, 1993). However, hydrological models, physically-based or lumped, always
contain simplifications and approximations of the real environment as it is impossible to characterise hydrological systems
with enough detail to simulate all the processes and features that they include (Clark et al., 2017; Freeze and Harlan, 1969).
As outlined in Clark et al., (2017), it is imperative that in order to improve the physical realism and applicability of
physically-based models, focus on parameter estimation is necessary.

Gupta et al., (2014) considered hydrological studies using at least 30 catchments to be "large-sample hydrology". Large-
sample hydrological studies assess the use of hydrological models on many catchments with varying characteristics (Lane et
al., 2019). The assessment of a large number of catchments allows the performance of the model to be evaluated over a range
of different climatic, land-surface, topographical, geological and other catchment characteristics. This approach also informs
about the suitability of a model for simulating different types of catchment characteristics, it can help identify spatial patterns
and areas where the model is unable to produce satisfactory results, and can highlight key processes that should be improved
(Lane et al., 2019). Large-sample hydrology can also help identify appropriate model parameter values for different
catchment characteristics and can ultimately help to produce parameter libraries which can be used to improve poorly gauged
catchments or simulate ungauged catchments (Perrin et al., 2008; Rojas-Serna et al., 2016). In addition, large sample
hydrology studies support and facilitate a better understanding of the amount of uncertainty that can be expected in a model
(Gupta et al., 2014). In the UK, national scale large-sample hydrological studies are being increasingly applied (Coxon et al.,
2019; Lane et al., 2019; Lees et al., 2021; Smith et al., 2024), facilitated by an increase in computing power, in addition to
the development of quality controlled hydrological modelling datasets which contain large numbers of catchments. For
example, the Catchment Attribute and MEteorology for Large-Sample studies (CAMELS) dataset which includes full
meteorological data, discharge data, catchment masks, Digital Elevation Model (DEM) data, soil data and land-use data



together with a range of other catchment attributes for 671 catchments in Great Britain (Coxon et al., 2020). These types of datasets are an important development in hydrological modelling as they provide a consistent set of data which enables the

results from different hydrological models to be compared. However, national hydrological studies in the UK have been limited to the use of conceptual and data-driven models, despite the advancement in data availability and computing power. Only two studies have applied a physically-based model to a national, large sample study of UK catchments (Lewis et al., 2018b; Smith et al., 2024). Complementing Smith et al., (2024), this paper provides detailed background of that study, and presents subsequent developments in the modelling approach.


Across the globe it is a similar story, with national scale hydrological studies being facilitated through the increase in availability of quality-controlled hydrological datasets for many countries (e.g. USA: Addor et al., 2017; Chile: Alvarez-Garreton et al., 2018; Australia: Fowler et al., 2021; Switzerland: Höge et al., 2023). These large-sample hydrological studies again primarily use conceptual and data-driven machine learning models. For example, in France 237 catchments were

modelled using 13 lumped conceptual models (van Esse et al., 2013), and in another study 1061 French catchments were modelled using 17 lumped conceptual hydrological models (Velázquez et al., 2010). Similarly, conceptual models were applied nationally in Finland at 67 study sites (Veijalainen et al., 2010), and across the whole of New-Zealand (McMillan et al., 2016). However, over the past 30 years, a national water resources model of Denmark (DK-model) has been developed and updated several times (Henriksen et al., 2003; Højberg et al., 2013). It uses a physically-based, spatially distributed

modelling system, MIKE-SHE, which covers the entire land phase of the hydrological cycle. In addition, a coupled physically-based model, Parflow-CLM was implemented in Europe at a 3km resolution (Naz et al., 2023) and Parflow CONUS 2.0 across the United States at a 1km resolution (Yang et al., 2023).

Data-driven, conceptual, and physically-based models are all able to be used for practical purposes such as predicting stream

or river flows under normal conditions. However, if conditions shift beyond the range of prior experience (e.g. longer droughts due to climate change), evaluation that the models are capable of correctly simulating the hydrological processes under these new conditions, is needed. The confidence that they are able to do this is increased if the model is "getting the right results for the right reasons" (Kirchner, 2006). However, in hydrological simulations that use the CAMELS large-sample hydrology dataset, model analysis only considered the simulation of the outlet discharge and does not consider the

physically realism of the parameters produced by the models.

Consequently, the four objectives of this research are to:

     1)      Run SHETRAN, a physically-based hydrological model, for 668 CAMELS-GB catchments, in addition to 30 catchments in Northern Ireland, using measured and standard library parameter values. Subsequently

130                referred to as: 'Uncalibrated-SHETRAN-UK'.



2)  Automatically calibrate SHETRAN for the same 698 catchments to reduce the manual time required by a user to set-up, calibrate and validate a model, and improve model performance through the better optimisation of parameters. Subsequently referred to as: 'Autocalibrated-SHETRAN-UK'.

3)  Compare the results of simulated outlet discharges from Autocalibrated-SHETRAN-UK with simpler conceptual and data driven models.

4)  Consider the physical realism of the parameters obtained during the autocalibration process to investigate and increase confidence that the model is performing well.

## 2 Data and Models

The following sections provide background on the study area, existing SHETRAN setup, and automatic calibration process.

### 2.1 Study Area

This study modelled 698 catchments across the UK. The majority of catchments, 668, were chosen due to their inclusion in the CAMELS-GB dataset – the first large-sample catchment hydrology dataset for Great Britain. These catchments cover a wide range of climatic, hydrological, landscape, and human management characteristics across Great Britain. The dataset includes daily time-series from 1970 to 2015 for a range of hydro-meteorological variables for example rainfall, potential evaporation, temperature, and river flow, in addition to a comprehensive set of catchment attributes such as topography, landcover, soils, and hydrogeology. Three catchments included in the CAMELS-GB dataset, 39061, 76011, and 80005, were omitted from this study as they were too small for the purposes of this research. In addition to the 668 catchments in Great Britain, 30 catchments from Northern Ireland were also included. These were not quality controlled; however, they represent all available catchments that are entirely within Northern Ireland as there were data constraints for any crossing the border with the Republic of Ireland.

### 2.2 SHETRAN Hydrological Model

SHETRAN, a physically-based, spatially-distributed model (Ewen et al., 2000) is used within this study as it has proven to be valuable in the modelling of both surface water and groundwater as well as other applications, from assessing the effect of deforestation on peak flows and sediment concentrations (Birkinshaw et al., 2011), to more groundwater-based studies such as recharge estimation (Walker et al., 2019). The most convenient way of visualising SHETRAN is as a set of vertical columns with each column divided into finite-difference cells. The lower cells in each column contain aquifer materials and groundwater, higher cells contain soil and soil water, and the uppermost cells contain surface waters and the vegetation canopy. The grid of columns also contains information on the topography of the catchment, with channels specified around the edge of the finite-difference cells. SHETRAN is capable of representing fully integrated surface-groundwater systems and allows interactions between rivers and aquifers to occur. Within this work, the meteorological inputs and hydrological



outputs used for calibration and validation are considered on daily timescales, however, SHETRAN uses adaptive timesteps that are at a higher resolution for calculations. SHETRAN requires a range of data in order to be run. This is described in detail in Lewis et al., (2018b).

In addition to case study applications, SHETRAN has been implemented at a large scale, with Lewis (2016) applying it to 306 catchments across Great Britian (SHETRAN-GB), where it broadly demonstrated a strong ability to simulate river flows. Building upon this foundation, this research will be expanded to include catchments from Northern Ireland and CAMELS-GB catchments, resulting in a total of 698 catchments being modelled across the United Kingdom (UK). This extended modelling framework will henceforth be referred to as SHETRAN-UK.


Of the 698 catchments, SHETRAN-UK was run for 682 catchments at a 1km grid resolution, i.e. the catchment is split into 1km-by-1km vertical columns. 16 larger catchments with catchment areas greater than 2000km$^2$ were run at a 5km resolution, see Appendix A for details. The uncalibrated version of SHETRAN is based on the work in Lewis et al., (2018b), which was shown to simulate river flows with reasonable accuracy (Seibert et al., 2018). Uncalibrated-SHETRAN-UK used

spatially variable soil and aquifer properties with seven land use classes. Appendix B presents the assigned physically realistic parameter values for the uncalibrated-SHETRAN-UK.

**2.3 SHETRAN Automatic Calibration**

The development of the automated calibration method of running SHETRAN models aims to reduce the amount of manual time required to set-up, calibrate, and validate the model, with the intention of also improving model performance through

better optimisation of parameters. In order to reduce the computational demands of the automatic calibration process, three simplifications to the model were needed:

1. A single homogeneous shallow soil type was applied across each catchment with a uniform thickness that is varied (calibrated) during the autocalibration process.
2. A single deep soil (aquifer) unit was used across each catchment with a fixed thickness of 20 metres.

3. Only two land use types were used: urban and rural.

The autocalibration process calibrated the following five parameters selected based on the findings of previous sensitivity studies (Op de Hipt et al., 2017; Sreedevi et al., 2019):

1. Deep soil (aquifer) conductivity – controls the rate at which water flows through the deeper soil aquifer.
2. Shallow soil conductivity – controls the rate at which water flows through the soil layers in a catchment.
3. Shallow soil depth – controls the vertical thickness of the soil cells in each column.
4. AE/PE Ratio – actual evaporation divided by the potential evaporation rate when the soils are at field capacity. This parameter also corrects errors in the overall water balance of a catchments, relating to, for example, abstractions,
effluent return, or different surface water-groundwater catchment boundaries.



5. Urban precipitation fraction – this parameter splits urban precipitation into two fractions as a percentage; one where precipitation flows directly into stormwater drains, and one where precipitation is removed due to flowing in combined sewers. This based on some of the techniques developed in (Birkinshaw et al., 2021) and is implemented within the SHETRAN executable itself.


The bounds that were defined for each of these calibrated parameters are presented in Appendix C in addition to the non-calibrated parameters.

The autocalibration process of SHETRAN is based on the Shuffled Complex Evolution method, developed at the University
of Arizona (Duan et al., 1992, 1994, 1993). This method has been proven to be successful in effectively and efficiently calibrating watershed models through numerous studies, providing superior performance compared to alternative optimisation algorithms (Zhang et al., 2015). For most catchments, per autocalibration process, 462 simulations were carried out. For details of the autocalibration process, see Appendix D.

## 2.4 Model Calibration and Validation

The automatic calibration of each catchment is based on 30 years of data. To ensure that all models were fully 'spun up' prior to any calibration attempts, the models were run using a 10-year initial period of 1/1/1980 – 31/12/1989. Then, a standard split-sample test was used (Klemeš, 1986), with a 10-year calibration period from 1/1/1990 – 31/12/1999, and a 10-year validation period from 1/1/2000 – 31/12/2009.

Model performance was evaluated based on daily observational discharge data at the outlet of each catchment. The main metric calculated for each simulation was the Nash-Sutcliffe Efficiency (NSE) (Nash and Sutcliffe, 1970). The range of NSE values lies between 1.0 (perfect correspondence between simulated to observed) and -∞. An NSE of zero indicates that the model simulations have the same explanatory power as the mean of the observations (Knoben et al., 2019).

NSE was chosen due to its broad usage in the hydrological community, making it well suited for benchmarking results and allowing the direct comparison of results with other national scale catchment modelling studies, for example (Coxon et al., 2019; Lane et al., 2019; Lees et al., 2021). The largest disadvantage of the Nash-Sutcliffe Efficiency emerges through the calculation of the differences between observed and simulated values as squared values. This means the calibration focuses on high flows and the performance of the model during low flows is often overlooked (Krause et al., 2005). Consequently, a
good NSE value doesn't necessarily indicate that low flows are simulated well within a model. Thus, five other metrics were calculated for each simulation: Percentage Bias (PBias), LogNSE, Kling-Gupta Efficiency (KGE), Inverse KGE, and the Mean of KGE and Inverse KGE (Garcia et al., 2017; Gupta et al., 2009; Moriasi et al., 2007). PBias has a range between +∞ and -∞. The other metrics range between 1.0 (perfect fit of simulated to observed) and -∞.



## 2.5 Model Comparison

Model comparison allows the strengths and limitations of the models to be understood, allows the uncertainty associated with the model to be recognised, and helps to improve confidence in the performance of the models (Lane et al., 2019). Consequently, Autocalibrated-SHETRAN-UK was compared to the performance of several other national scale studies, which use conceptual and data-driven models (Hannaford et al., 2023; Lane et al., 2019; Lees et al., 2021). However, the comparison is challenging because although each study is based on the 671 CAMEL-GB catchments, different subsets of catchments are used; for instance, this study simulates 668 catchments, whereas others use much smaller samples. Additionally, variations in calibration and evaluation time periods further complicate comparisons. Furthermore, while some studies use a split-sample calibration-validation approach (out-of-sample evaluation) (Lees et al., 2021), others calibrate and evaluate performance within the same time period (in-sample evaluation) (Hannaford et al., 2023). Despite these differences, all studies use NSE as the performance criterion for calibration and validation, allowing for a degree of comparison with SHETRAN-UK.

Firstly, the results of Autocalibrated-SHETRAN-UK were compared to the best results from the Enhanced Future Flows and Groundwater project (eFLaG, Hannaford et al., 2023), which used 4 models: G2G, GR4J, GR6J, and PDM to model 200 catchments nationally in the UK. Results were also be compared against two data-driven models, a long short-term memory (LSTM) model and the Entity Aware LSTM (EALSTM) model, which were used in a benchmarking study by Lees et al., (2021). Finally, it was compared against the highest performing model from Lane et al., (2019), which used the lumped, conceptual models TopModel, ARNO/VIC, PRMS, and Sacramento, for the modelling of 1000 catchments nationally.

## 3 Results

### 3.1 Uncalibrated-SHETRAN-UK model performance

The Uncalibrated-SHETRAN-UK simulations (which use default parameter values, a two layer spatially heterogeneous subsurface and spatially varied land use) produce NSE values ranging from -108.74 to 0.94 for the validation period (Fig. 1a, Fig. 2). Nationally, for the validation period, 48% of the 698 catchments return a NSE score of ≥ 0.7, 20% of catchments receive scores between 0.5 and 0.7, 15% of catchments between 0 and 0.5, and 17% of catchments produce NSEs of less than 0. The median NSE score of Uncalibrated-SHETRAN-UK is 0.69. In these uncalibrated simulations, SHETRAN performs well in the north and west of the UK, where the catchments are surface water dominated and have low base flows. Generally, Uncalibrated-SHETRAN-UK performs poorly in the south-east region of the UK where there is a Chalk aquifer and groundwater flow dominates, as shown by the diagonal band of dark blue stretching from the south coast into East Anglia in Fig. 1c.




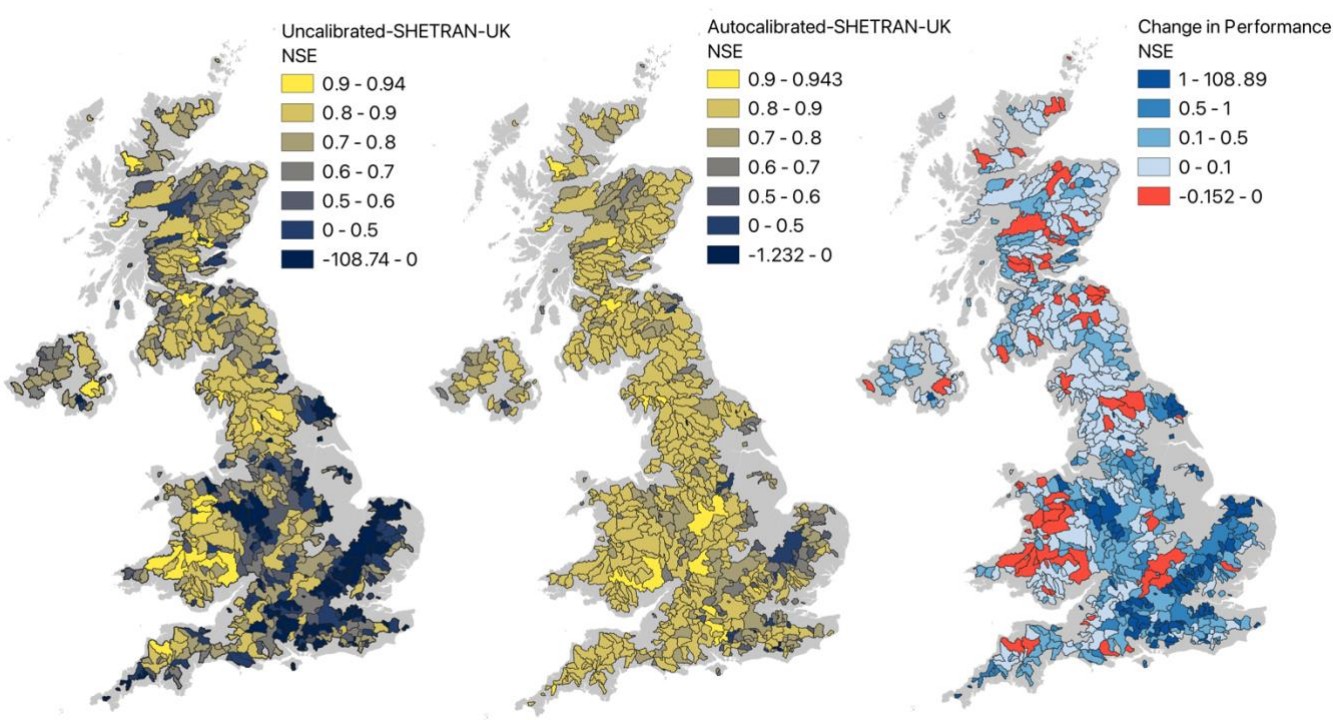

**Figure 1. Catchment map showing the NSE values for the validation period (2000-2009 inclusive) for all 698 catchments a) Uncalibrated-SHETRAN-UK, b) Autocalibrated-SHETRAN-UK, c) Change in model performance between Uncalibrated-SHETRAN-UK and Autocalibrated-SHETRAN-UK.**





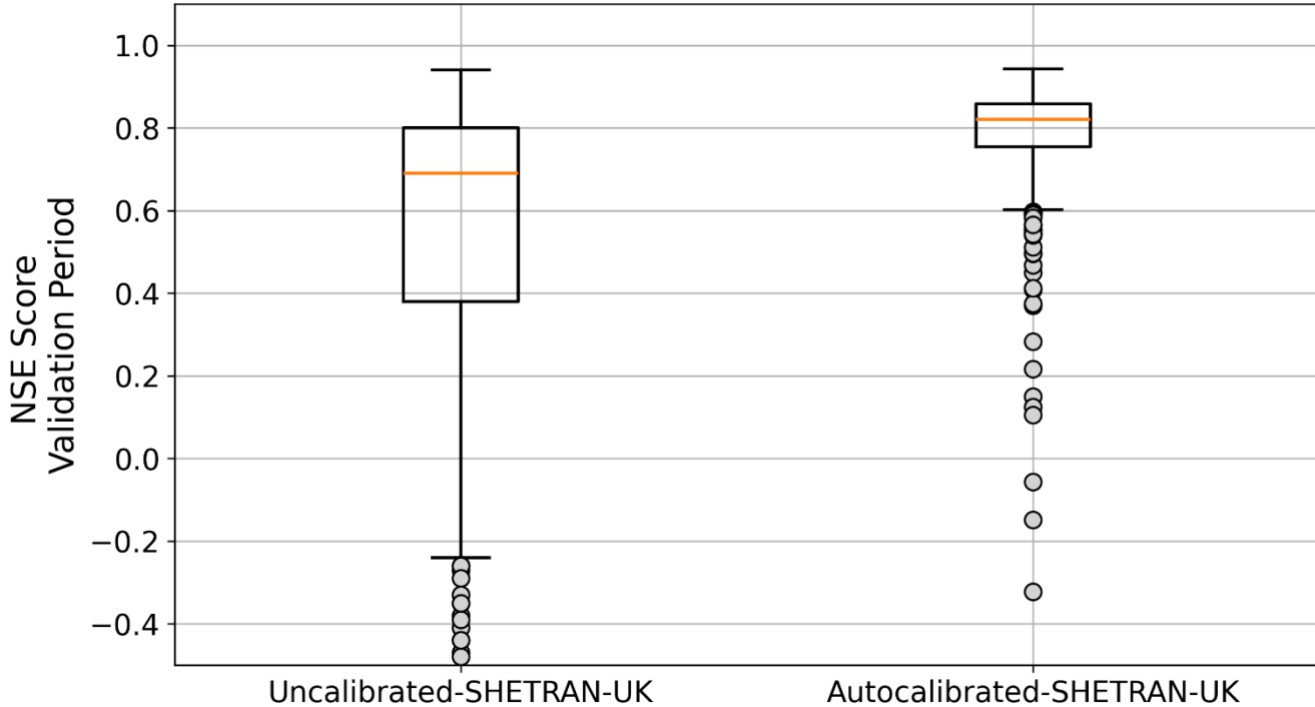


**Figure 2. Box plot showing the overall model performance for the validation period (2000-2009 inclusive) for all 698 catchments for the Uncalibrated-SHETRAN-UK and Autocalibrated-SHETRAN-UK.**

## 3.2 Autocalibrated-SHETRAN-UK model performance

After the autocalibration process, NSE scores for the validation period range nationally from -1.2 to 0.94 (Fig. 1b). The national median NSE score is improved from 0.69 to 0.82 and 85% of catchments have an NSE score of ≥ 0.7. Only 18 catchments have an NSE score of ≤ 0.5, with 5 of these receiving scores of less than 0. Figure 2 shows the interquartile range of SHETRAN is reduced from 0.42 to 0.10 after the autocalibration, indicating that there is less variability in the performance of catchments. Overall, results show that applying the autocalibration process significantly improves model

performance nationally, particularly in catchments that were found to perform poorly in uncalibrated-SHETRAN-UK.

Figure 1c shows the difference in NSE values between the uncalibrated and automatically calibrated SHETRAN setups. Red indicates that after autocalibration the surface water of a catchment is modelled less accurately than with uncalibrated SHETRAN, with a decrease in NSE score. Performance improved in 623 catchments (90%), as shown by the blue

catchments in Fig. 1c. The autocalibration process has the most significant improvement in the NSE score for catchments in the south-east regions of the UK that are underlain by Chalk, in addition to catchments near the Welsh border, which were




largely simulated poorly by Uncalibrated-SHETRAN-UK. In Section 3.2.2, it is suggested that there is no correlation between model performance and Baseflow Index. Consequently, this could suggest that Autocalibrated-SHETRAN-UK is equally good at simulating surface water-dominant and groundwater-dominant catchments, and the significant improvements

in the performance of groundwater-dominated catchments is because they were simulated poorly in Uncalibrated-SHETRAN-UK, and hence show the greatest increase through the autocalibration process.

Results from four alternative objective functions are shown in Fig. 3 comparing the autocalibrated flows to the observed flows. All show high mean values, indicating that all aspects of the flow regime are able to be simulated well (including low

flows).

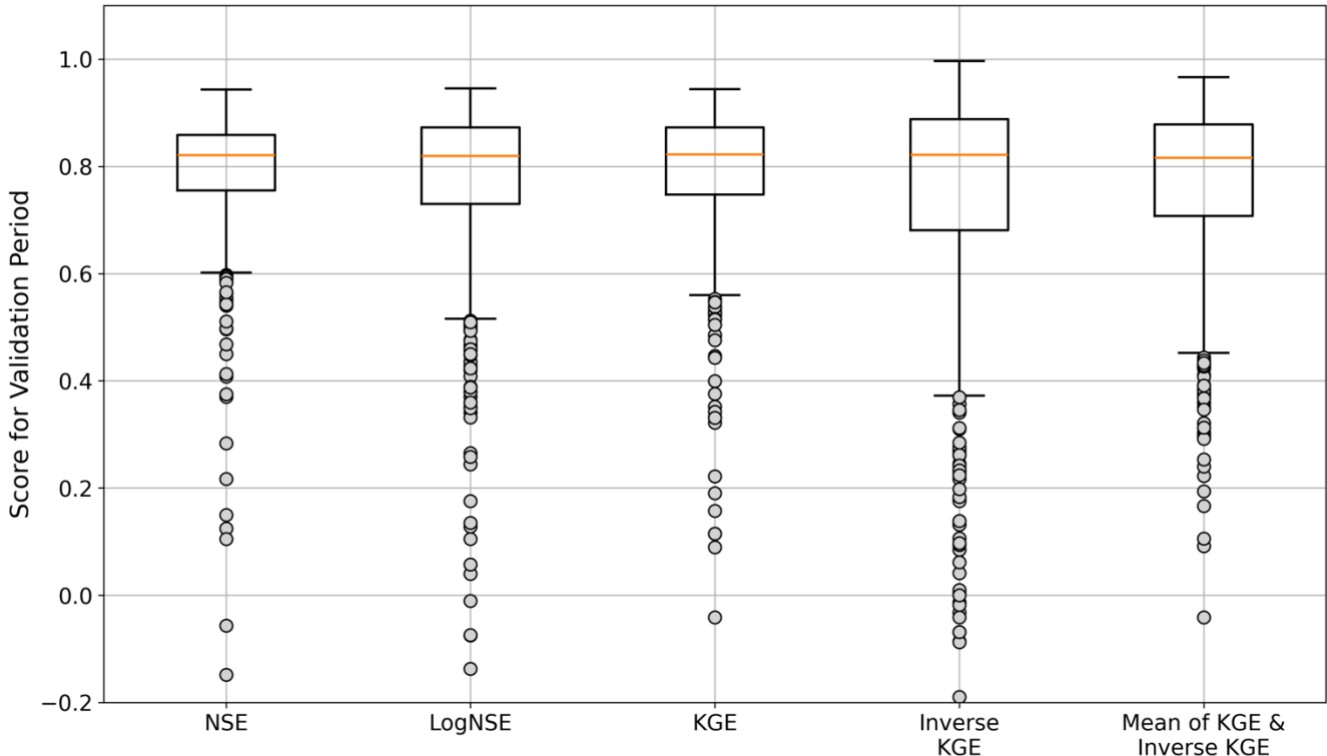

**Figure 3. Box plot for five different objective functions calculated for Autocalibrated-SHETRAN-UK, showing the model performance for the validation period (2000-2009 inclusive) for all 698 catchments.**

190 Autocalibrated-SHETRAN-UK catchments have an NSE score of greater than 0.8, and a PBIAS of between -5% and +5% and therefore are considered to be the most accurately modelled catchments by Autocalibrated-SHETRAN-UK. When underlain by the BGS hydrogeological aquifer map, as shown in Fig. 4, it is clear that the majority of these catchments are located in areas of low or moderately productive aquifers, and generally are not located in areas where there are highly productive aquifers. This suggests that although autocalibration improves performance in a range of different catchment



types including catchments underlain by highly productive aquifers (e.g. the Chalk) as discussed above in Fig. 1, it still does not model these catchments as accurately as it models catchments in areas that are underlain by less productive aquifers.

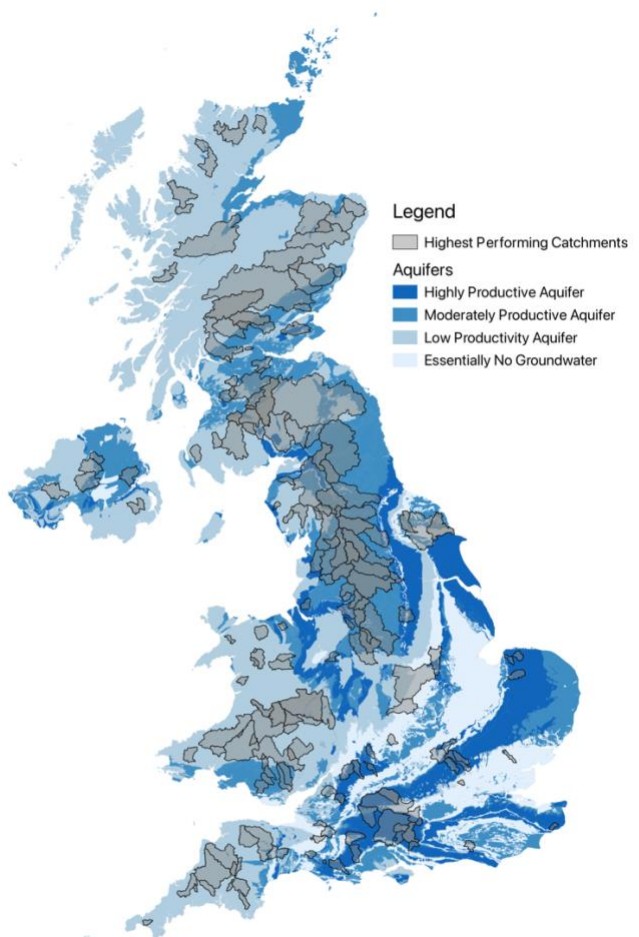

**Figure 4. Map showing the 190 highest performing Autocalibrated-SHETRAN-UK catchments underlain by the BGS**
**hydrogeological aquifer map (BGS, 2020).**

### 3.2.1 Comparisons of performance in calibration and validation periods

When using a split-sample approach for calibration and validation, catchments are generally considered well-performing if they exhibit similar performance in both periods, or a slightly lower performance in the validation period. This is expected,
as the validation period is subject to different meteorological conditions and therefore leads to a slightly different hydrological response compared to the calibration period. Figure 5 illustrates that, although some variability exists between the calibration and validation periods across catchments, Autocalibrated-SHETRAN-UK generally has similar performance





in both periods, with the NSE in 95% of catchments varying by no more than ±0.1. This suggests that the model is simulating catchment processes well and that the autocalibration process is optimising parameters that allow for the
production of good simulations under different meteorological conditions to which they were calibrated.

However, 30 catchments exhibit a notable decrease in model performance between the calibration and validation periods, with some NSE reductions decreasing by 0.75. The majority of these catchments are located in the south-east of the UK (Fig. 5c) and experience considerable groundwater abstractions, along with changes in the abstraction rates between the
calibration and validation periods, consequently resulting in increased or decreased observed river flows during the validation period. Abstractions are not incorporated into the model and consequently could possibly explain the corresponding reduction in the NSE score for these catchments.

Overall, the comparison suggests that the calibration process is performing correctly, producing good parameter values, and
allowing SHETRAN to simulate catchments under meteorological conditions of a period for which they were not calibrated to.

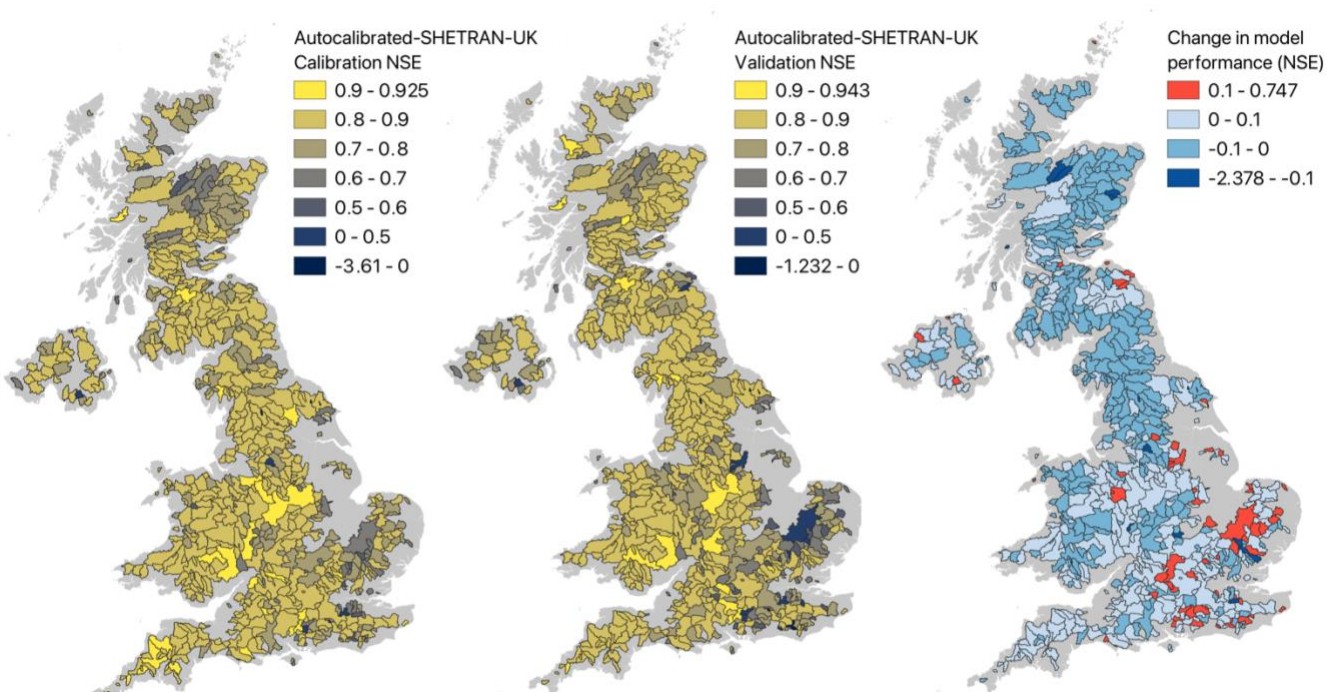

**Figure 5. Catchment map showing the NSE values for all 698 catchments a) calibration period (1990-1999 inclusive), b) validation period (2000-2009 inclusive) c) difference between NSE values for the calibration and validation periods (red shows significantly**
**worse in validation).**




### 3.2.2 The effect of different catchment characteristics on model performance

The performance of the autocalibration, measured using NSE, was evaluated against catchment descriptors to identify whether specific catchment characteristics were associated with poorer model performance. The catchment descriptors considered are: area, base flow index (BFI), elevation 50th percentile, and urban fraction percentage (Fig. 6). These

descriptors were selected from the UK National River Flow Archive (NRFA) using the NRFA API (NRFA, 2025) to ensure analysis across a diverse range of catchment types. The BFI (Gustard et al., 1992) is a measure of the proportion of the river runoff that derives from stored sources. It ranges between 0 and 1 and the higher the value, the higher the groundwater component in the catchment.

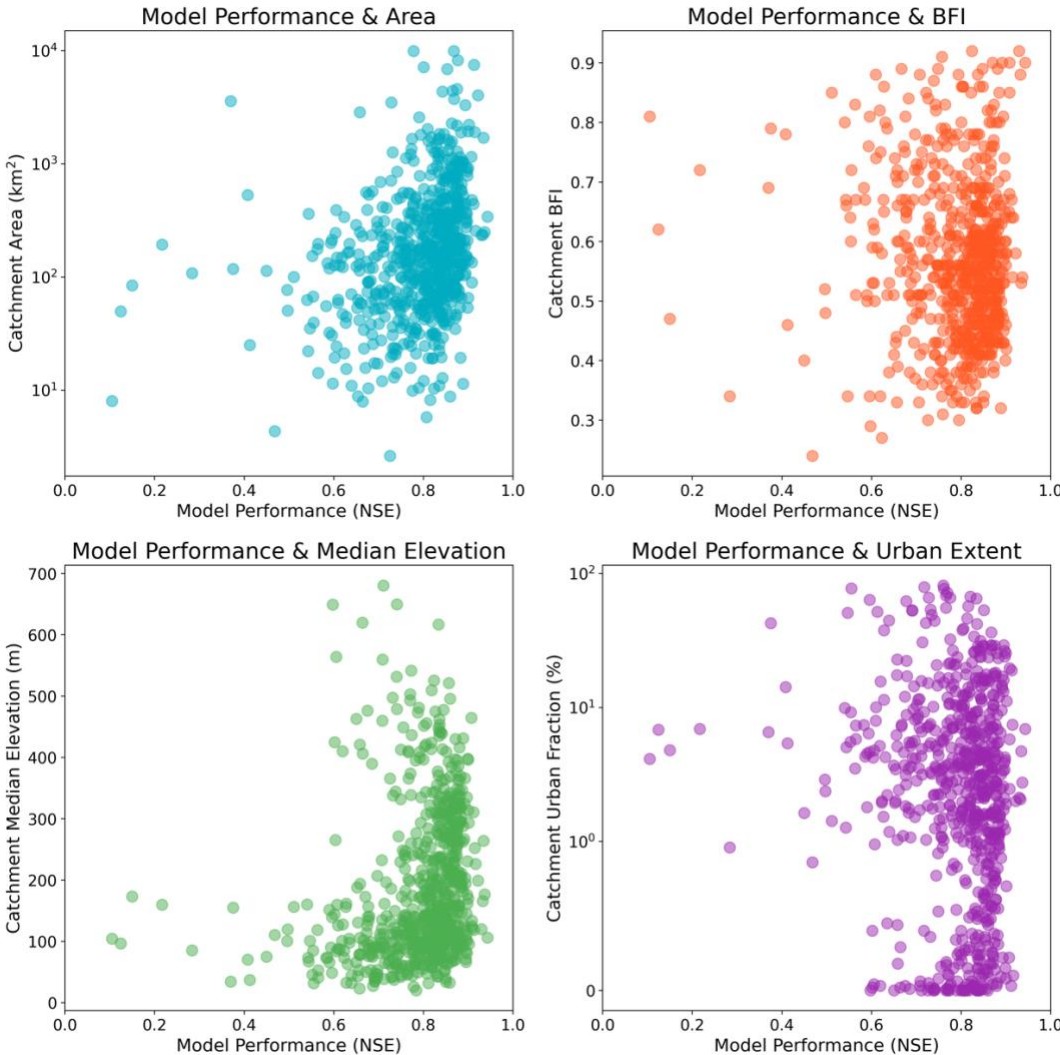

**Figure 6. Scatter plots showing model performance for different catchment descriptors (a) Area, (b) BFI, (c) Median Elevation, (d) Urban Extent.**



There is no clear relationship between catchment area and model performance (Fig. 6a), however, the poorest performing catchments are typically catchments with small areas. This may be due to catchment processes becoming smoothed over the larger catchments, allowing for easier calibrations.

The performance of Autocalibrated-SHETRAN-UK does not appear to be influenced by the BFI score of any given catchment (Fig. 6b), with no clear relationship between the two. This indicates that SHETRAN is performing well for both surface and groundwater dominated catchments.

Elevation also does not seem to have a significant impact on how well a catchment is simulated by Autocalibrated-SHETRAN-UK (Fig. 6c). Snow accumulation and snow melt are important processes in high elevation catchments; therefore, this finding suggests that these processes are reasonably well simulated.

Finally, there does not appear to be a relationship between the percentage of urban cover in a catchment and the performance of the model (Fig. 6d). However, it is good to note that Autocalibrated-SHETRAN-UK generally performs well in catchments that have high urban extent percentages, in addition to performing well in more natural catchments. Consequently, it is clear that Autocalibrated-SHETRAN-UK performs well for a range of different catchment types.

### 3.2.4 Model performance in contrasting catchments

In this section, the accuracy of Autocalibrated-SHETRAN-UK to correctly simulate rivers flows is assessed in detail for two randomly selected catchments with contrasting climatic, hydrological, and landscape characteristics. One groundwater dominated catchment, Stringside at Whitebridge (NRFA catchment number: 33029) located in the east of England, and one surface water dominated catchment, Lune at Lunes Bridge (NRFA catchment number: 72015) located in the north-west of England.

Catchment 33029, Stringside at Whitebridge, is located in East Anglia and has an area of 99km$^2$. Land use is predominately arable, with some grassland and woodland. Urban land cover accounts for 4% of the catchment area. The catchment is mostly low-lying, with elevations ranging from 7m to 83m, and it is predominately underlain by Chalk with around 27% of the catchment area overlain by highly impermeable superficial deposits. It has a baseflow index of 0.84 (NRFA, 2025) meaning that the river discharge in the catchment is significantly connected to groundwater systems.

Figure 7a shows this catchment was well simulated by SHETRAN, with a NSE of 0.78 for the validation period of 2000-2009 inclusive. The simulated flows follow the observed baseflow, but peak flows are typically higher than those simulated





by the model. This suggests that SHETRAN is simulating the groundwater-surface water interactions well but is not simulating run-off correctly following rainfall events. The model underpredicts these peaks due to challenges in accurately capturing rapid surface runoff contributions over the area of the catchment covered in superficial deposits. The autocalibration uses a simplified setup with the same soil and aquifer parameters used in each column. As the catchment discharge response is dominated by areas in which there is a Chalk aquifer but no superficial deposits, the autocalibration

process will produce suitable parameter values for these areas rather than those areas where superficial deposits are present which produce the peak flows. The consequence of this simplified setup is considered in more detail within the Discussion section (Section 4).

    Catchment 72015, Lune at Lunes Bridge, is a predominately rural catchment covering an area of 142km$^2$. It is generally

dominated by high relief topography, with elevations ranging from 163m to 675m. It has moderate and low permeability bedrock, which are overlain by some superficial deposits. The baseflow index of the catchment is 0.30 (NRFA, 2025), indicating it is a catchment that is dominated by surface runoff, with limited groundwater contributions to streamflow. Figure 7b shows this catchment was exceptionally well simulated by SHETRAN with a NSE value of 0.90 for the validation period of 2000-2009 inclusive. Overall, the simulated flow data for the catchment follows the observed data closely, generally

capturing the timing and magnitude of peak discharges and low flows. There are instances where observed flow peaks exceed the simulated values, however, there are known to be large errors in the measurements of precipitation and discharge during these large events, and therefore it is unclear if the errors are due to a modelling issue or a data issue. The simulated sharp peaks and rapid declines are characteristic of a flashy system and are expected given the catchment's physical characteristics. It is evident that SHETRAN accurately simulates the hydrological behaviours well in this catchment where

streamflow is dominated by surface water runoff.








**33029 - Groundwater Dominant**

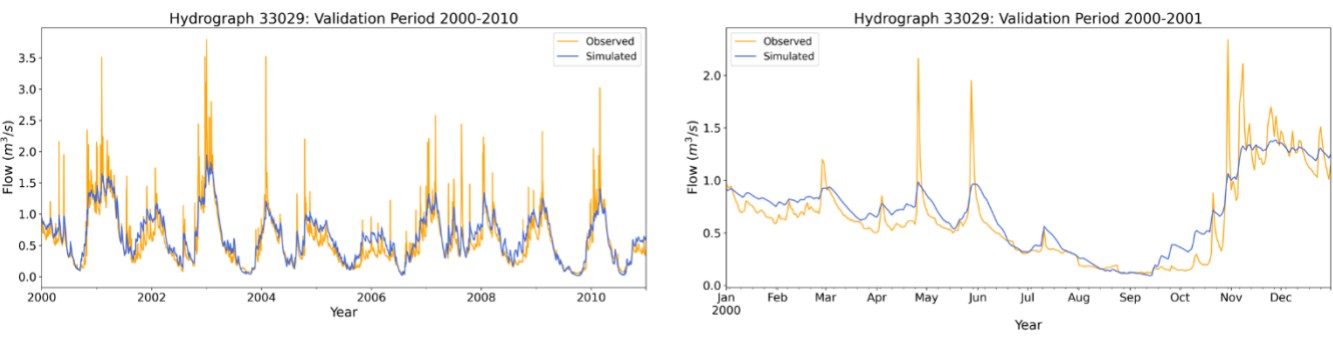

**72015 - Surface Water Dominant**

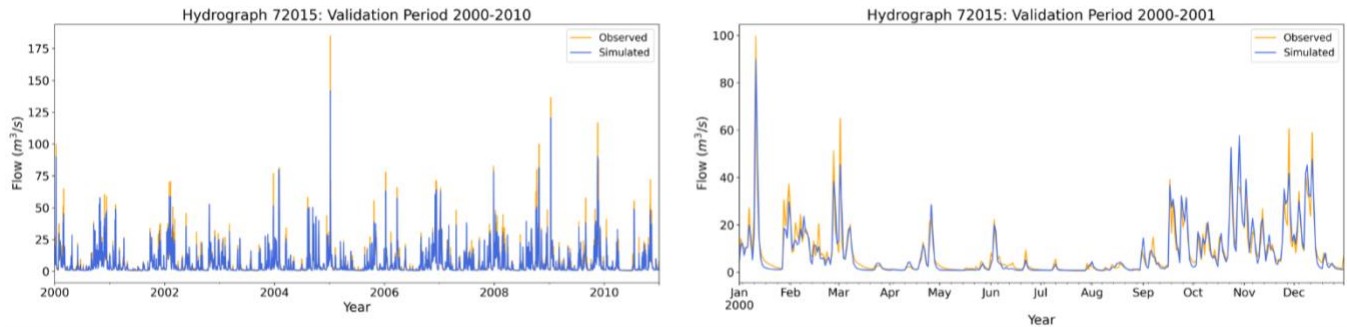

**Figure 7. Modelled and simulated outlet discharges for (a) catchment 33029 (Stringside at Whitebridge) and (b) catchment 72015**
**(Lune at Lunes Bridge).**

## 3.3 Performance of Autocalibrated-SHETRAN-UK compared to simpler conceptual and data driven models

The eFLaG project (Hannaford et al., 2023), simulated 200 catchments using four conceptual models; these produced NSE
scores between 0.85 and 0.86 for the GR4J, GR6J and PDM models, and a slightly lower NSE score for the G2G model. For
the same subset of 200 catchments modelled in the eFLaG study, Autocalibrated-SHETRAN-UK had a very similar
validated NSE score of 0.85. This suggests that the performance of Autocalibrated-SHETRAN-UK, is very comparable to
the performance of Hannaford et al., (2023).



Lees et al., (2021) simulated the 669 of the CAMELS-GB catchments using two data-driven models: LSTM and EALSTM

models. This produced median NSE scores of 0.86 and 0.88 respectively (Fig. 8), which are slightly higher than the

Autocalibrated-SHETRAN-UK median value of 0.82.

Lane et al. (2019) simulated over 1,000 catchments using four conceptual models: TopModel, ARNO/VIC, PRMS, and

Sacramento. The median NSE values for the subset of these 1000 catchments which were simulated by Autocalibrated-

SHETRAN-UK, were between 0.76 and 0.79 for the four conceptual models (Fig. 8). Consequently, this means that

Autocalibrated-SHETRAN-UK outperforms the four conceptual models used in Lane et al., (2019) when modelling most

catchments.

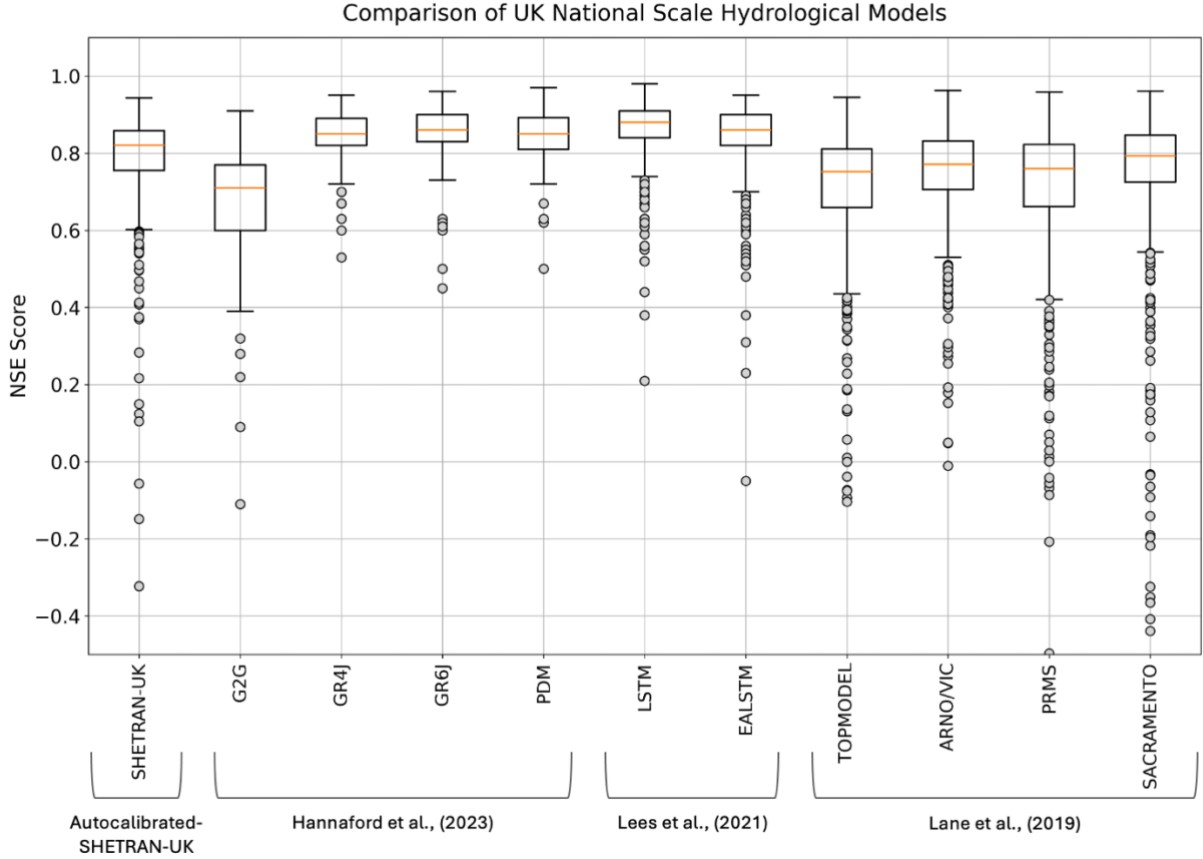

**Figure 8. Box plots showing the model performance of Autocalibrated-SHETRAN-UK and the models used in Hannaford et al.,**
**(2023), Lees et al., (2021), and Lane et al., (2019) for all catchments in each study that correspond to the catchments used in**
**Autocalibrated-SHETRAN-UK. It must be noted when making comparisons between models that different catchment sample**
**sizes, years and calibration/validation methods were used in each study.**



**3.4 Physical Realism of Parameters Derived from Autocalibrated-SHETRAN-UK**

In this section, the physical realism of the parameters produced in the simulations are considered, in order to evaluate the performance of Autocalibrated-SHETRAN-UK. This focuses on the sub-surface parameters as these can be compared to measured national transmissivity and hydraulic conductivity values.

**3.4.1 National Transmissivity Comparison**

Transmissivity is the rate at which water is transmitted through an aquifer (Uliana, 2025). It is calculated as the product of
the average hydraulic conductivity and thickness of the saturated portion of an aquifer. For the simulation an approximate measure of the transmissivity for each catchment can be calculated by the following Eq. (1):

$$T = K_{1sat} \times b_1 + K_{2sat} \times b_2 \qquad\qquad (1)$$

Where T is equal to the transmissivity (m²/day), $K_{1sat}$ (m/day) is equal to the calibrated saturated conductivity in the shallow
soil layer, which has a depth of $b_1$ (m), and $K_{2sat}$ (m/day) is equal to the calibrated saturated conductivity in the deep soil/aquifer layer, which has a depth $b_2$ (m) which is equal to 20m.

Some error is introduced as the saturation depth of each soil column varies both spatially and temporally as the water table rises and falls. Consequently, as the soil column is not always fully saturated, the calculated transmissivity is an upper bound
of the modelled transmissivity. Nevertheless, this approximation is useful for identifying which catchments have higher and lower transmissivity values and how simulated Autocalibrated-SHETRAN-UK transmissivity values vary at a national scale.

The transmissivity of each catchment was calculated and the values presented on a log scale in Fig. 9a. A comparison of the simulated values from Autocalibrated-SHETRAN-UK with the BGS Aquifer Designation Zones (BGS, 2020) (Fig. 9b),
demonstrates that the model produces spatially coherent transmissivity values. Catchments with higher transmissivity values are predominantly located in regions designated as having highly productive aquifers, indicating that the model captures key spatial trends in transmissivity. Similarly, in regions designated as having low groundwater productivity, Autocalibrated-SHETRAN-UK simulates lower transmissivity values. However, since each catchment is assigned a single conductivity/transmissivity value across its entire spatial extent, local variabilities, where catchments may contain multiple
aquifer types, are not fully captured, leading to some blurring of these distinctions.



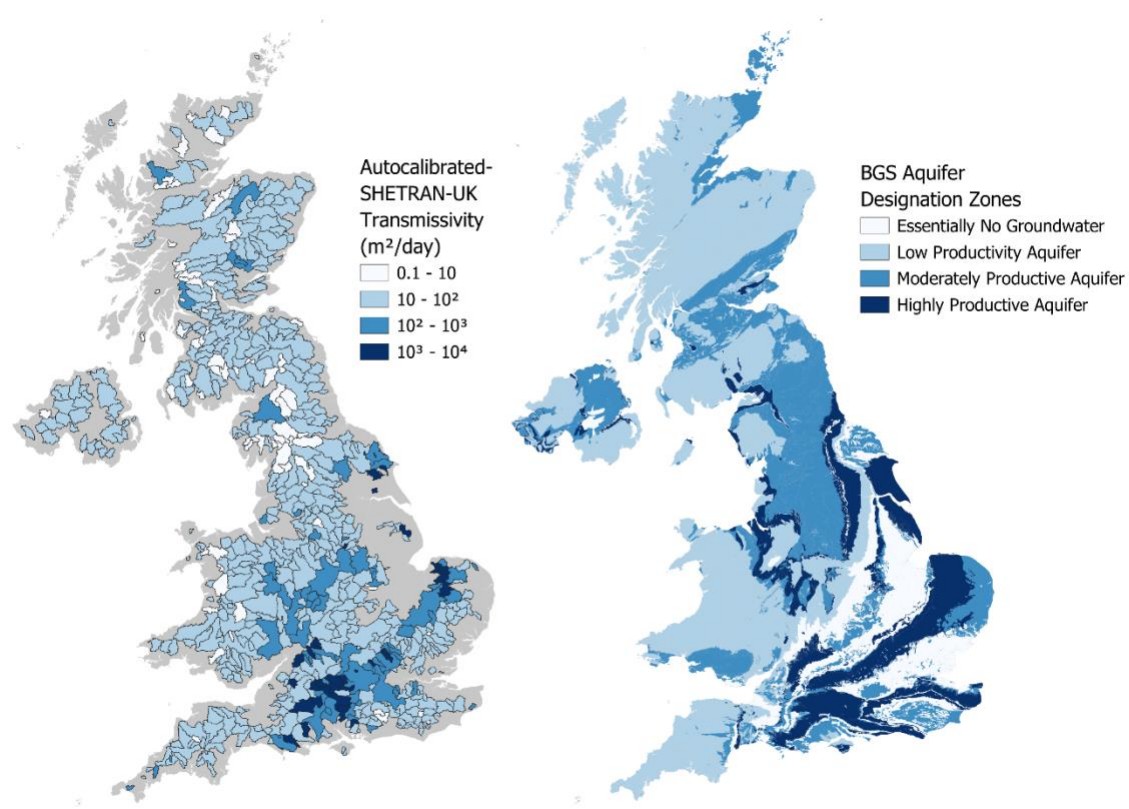

**Figure 9. a) Catchment map showing simulated catchment transmissivity values and b) a map showing the location of different aquifer zones (BGS, 2020).**

### 3.4.2 Chalk Aquifer Comparison

Chalk is the most important aquifer within Great Britain, mainly due to its occurrence in the south of England where

population density is higher and rainfall lower. In addition, it has a large outcrop area, allowing productive recharge (Allen et al., 1997). Despite its importance, the hydraulic properties of the Chalk are often poorly parameterised, resulting from a complex combination of matrix and fracture flow. This, combined with significant superficial drift deposits over the Chalk in certain areas, makes modelling catchments with Chalk aquifers challenging.

To assess whether Autocalibrated-SHETRAN-UK produces realistic parameter values, the transmissivity values were compared to those presented in 'The physical properties of major aquifers in England and Wales' (Allen et al., 1997), which compiled data from hundreds of groundwater investigations for sub-areas of aquifers. Figure 10 illustrates the sub-division in



the Chalk aquifer (Allen et al., 1997), with the regions and catchments considered in this part of the study highlighted in red. For each region, Fig. 11 provides the measured transmissivity values at the 25$^{th}$ percentile, median, and 75$^{th}$ percentile.

Generally, transmissivity values in Cambridgeshire are higher than those in Hertfordshire, East Suffolk and North Essex. The Cambridgeshire aquifer is largely free of superficial deposits, promoting the development of fractures and dissolution of the Chalk (Allen et al., 1997), thereby increasing the aquifer's transmissivity. In contrast, further south and east, more extensive and deeper superficial deposits cover the Chalk bedrock, partially confining the aquifer and consequently reducing its transmissivity due to less fracture development and dissolution of the Chalk.


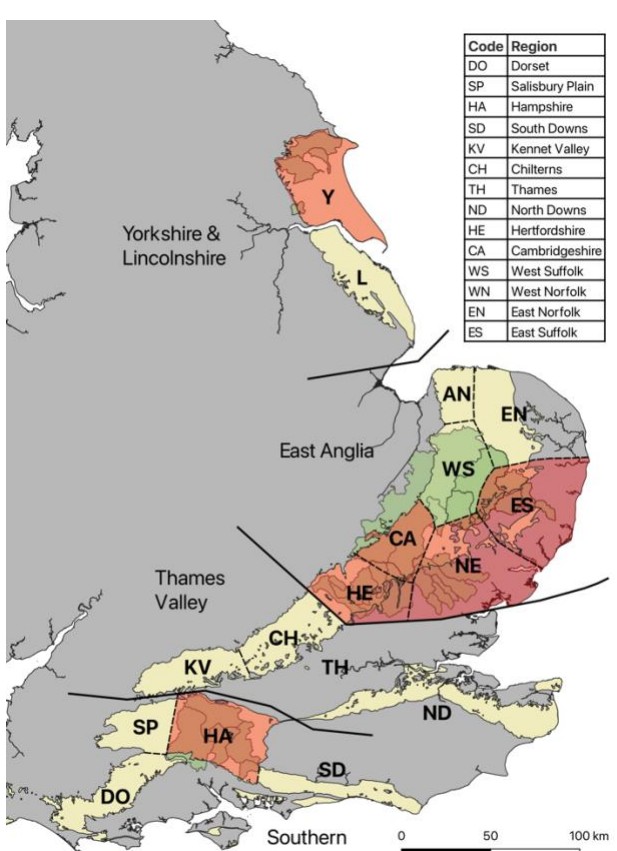

**Figure 10. Map showing Chalk aquifer regions, and the regions considered in this study (highlighted in red) with the associated catchments shown in green. Adapted from MacDonald and Allen (2001) .**

Figure 11 presents the Autocalibrated-SHETRAN-UK simulated transmissivity values for catchments located in the six comparison regions (see Fig. 10). In Yorkshire, Hampshire and Cambridgeshire the majority of simulated transmissivities fall within the interquartile range, with the exception of one catchment in Yorkshire and one in Hampshire, which are below



the 25[th] percentile of measured values. Overall, this suggests that the model is producing realistic transmissivity values in these regions.


Hertfordshire, East Suffolk and North Essex have lower measured transmissivity values compared to regions further north due to the semi-confined nature of the Chalk, where superficial deposits limit fracture development within the bedrock (Allen et al., 1997). The majority of Autocalibrated-SHETRAN-UK simulated transmissivity values for catchments in these regions fall below the measured 25[th] percentile. As the Chalk in these regions is covered by low permeability superficial

deposits, most of the discharge occurs though surface and shallow subsurface flow pathways, leading to rapid rises and falls in the hydrograph. The calibrated SHETRAN parameters reflect this response, with low transmissivities and water tables that are close to the ground surface. This is explored in more detail in the Discussion section.

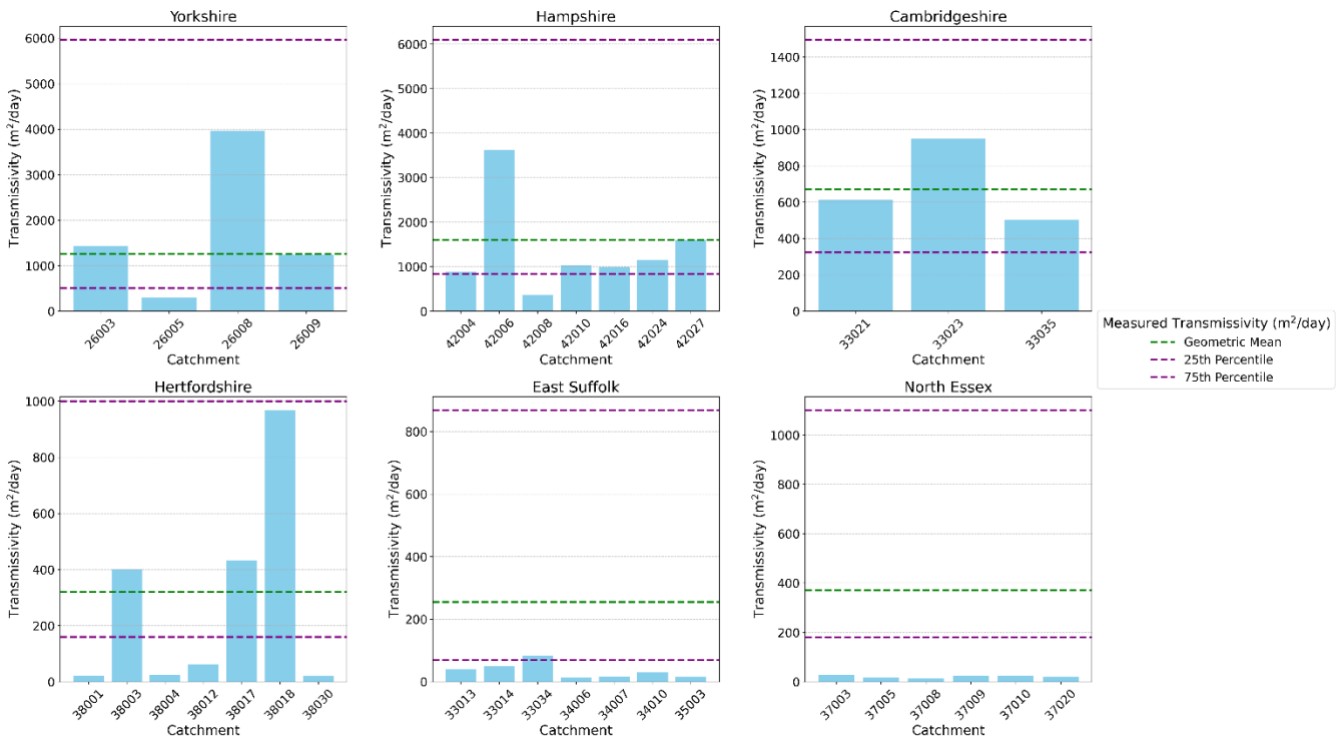

**Figure 11. Autocalibrated-SHETRAN-UK transmissivity values for catchments in selected regions plotted as a bar graph, with the corresponding quanitles found in Allen et al., (1997).**





### 3.4.3 Deep Aquifer Conductivity Comparisons

Figure 12 presents the UK aquifer conductivity values given in the Aquifer Properties Manual (APM) (Allen et al., 1997;
MacDonald and Allen, 2001) (see Lewis (2016) for more details), alongside the simulated conductivity values for
catchments in Autocalibrated-SHETRAN-UK. Nationally, APM conductivity values range from 0 to 130m/day. While the
conductivity values simulated by Autocalibrated-SHETRAN-UK generally align with these ranges, discrepancies remain. In
most cases, the model produces conductivity values similar to those measured within catchments, with lower values in the
north and west of the UK, and higher values in the south and east. However, in some catchments within the Chalk region of
eastern England, the autocalibration process does not yield realistic conductivity values. These catchments also exhibit lower
NSE scores which will be explored in Section 4. Nevertheless, when considering the transmissivity of the entire subsurface
in these catchments, the model appears to be behaving in a physically consistent way.

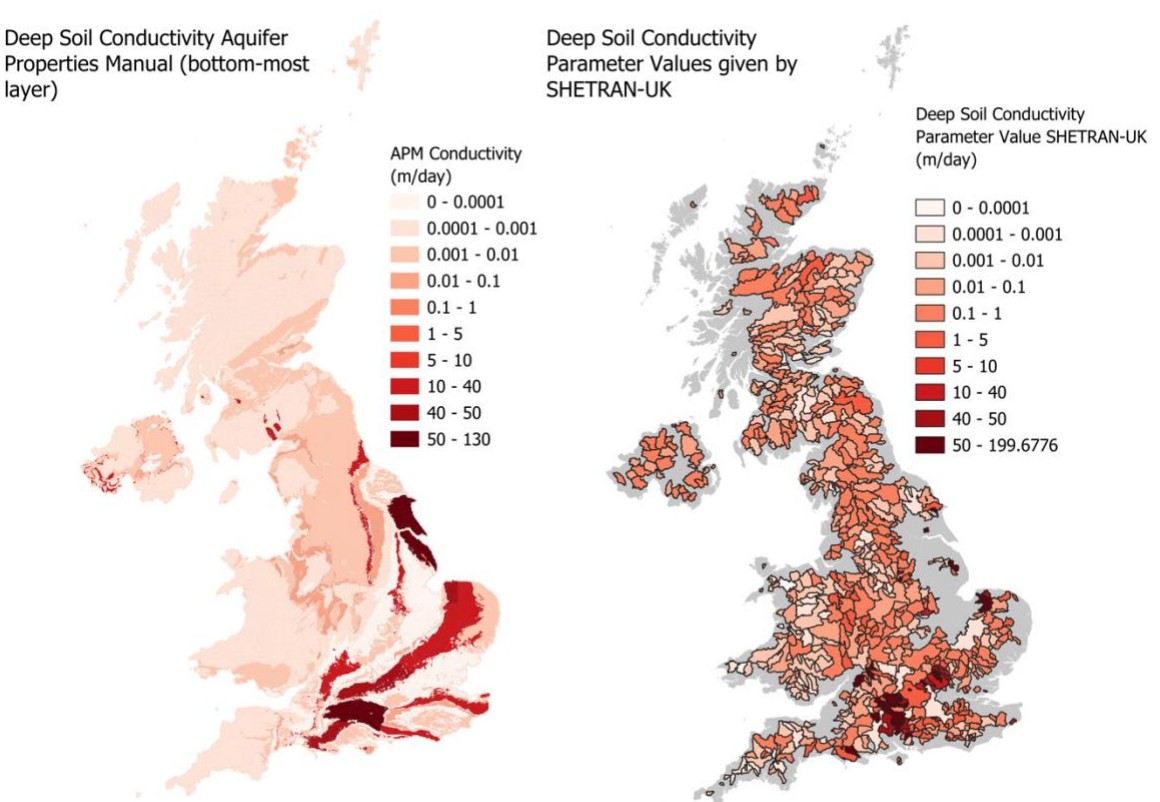

**Figure 12. a) Map showing measured conductivity values (Allen et al., 1997; MacDonald and Allen, 2001) and b) simulated deep soil conductivity values for each catchment from autocalibrated-SHETRAN-UK.**



## 4 Discussion

Autocalibrated-SHETRAN-UK simulations perform well for across a full range of flows, including both high and low
extremes, achieving high values across the five different performance metrics calculate. The performance of Autocalibrated-
SHETRAN-UK simulations in the validation period are comparable to those in the calibration period and the overall
performance of the model is similar to those produced using conceptual and data driven models. The model performs well in
both lowland and upland catchments, groundwater and surface water dominated catchments, urban and rural catchments, and
large and small catchments.


Analysis shows that the majority of poor Autocalibrated-SHETRAN-UK model performances occur in catchments with
considerable spatial variation in soil and aquifer properties. The simplifications in the autocalibration process of SHETRAN-
UK result in these catchments being modelled with a uniform soil and aquifer type in every soil column. This limitation is
evident in some East Anglian catchments (e.g., 33029, Section 3.2.4), which contain both highly permeable Chalk aquifers
and low-permeability bedrock, along with areas covered by low-conductivity superficial deposits. Figure 13 illustrates the
extent of both the Chalk aquifer and discontinuous low-permeability superficial deposits, which together create a complex
response in catchment hydrographs in East Anglia. High baseflows occur due to flow through the Chalk aquifer, while peaky
responses are superimposed on the high baseflow as a result of the superficial deposits. Seventeen catchments in this region
yield NSE validation values significantly lower than the national median (0.82), with values ranging from 0.50 to 0.73, as a
result of simplifications imposed by the autocalibration process on the subsurface structure. In pilot studies which have
incorporated the spatial variability and differences in bedrock and superficial geology found in these catchments to
SHETRAN, a significant improvement in the NSE is observed, for example in catchment 33029, the NSE value increases
from 0.78 to 0.9. Consequently, ongoing research aims to further integrate this spatial variability into the model to enhance
performance in these complex catchments.


Although integrating spatial variability in the subsurface of the model has the potential to improve simulation performance in
groundwater-dominant catchments, achieving an optimal balance between incorporating this variability and limiting the
number of calibrated parameters is crucial. In the uncalibrated model, these catchments can have up to one hundred
parameters associated with different soil and aquifer types. For a physically-based model like SHETRAN, automatically
calibrating such a large number of parameters becomes impractical due to computational constraints and could introduce
challenges related to parameter equifinality (Beven, 2006). Therefore, future calibration efforts will prioritise the use of
measured parameters wherever possible and will group soils and aquifers with similar properties to reduce complexity.



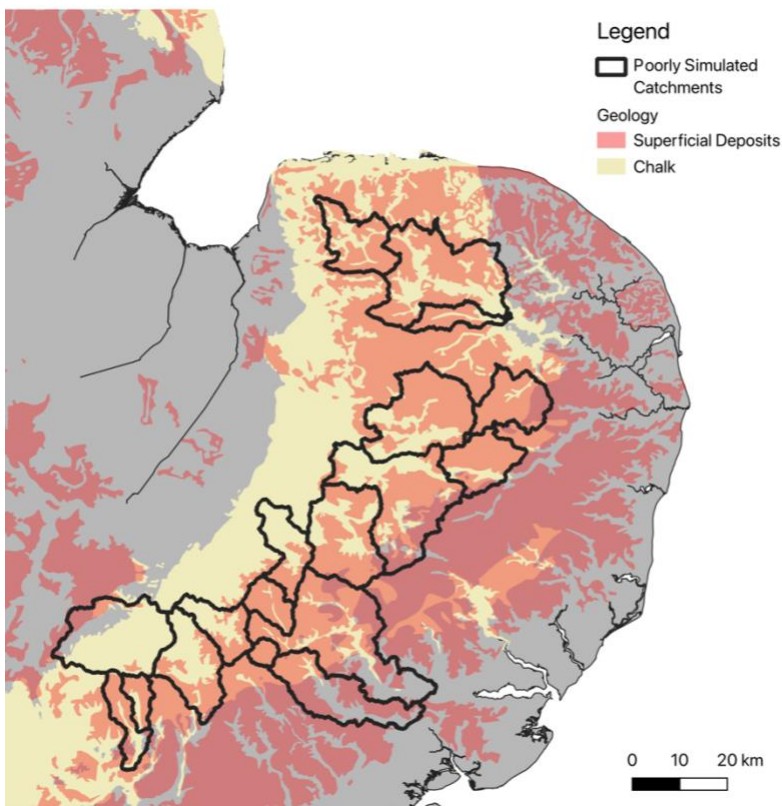

**Figure 13. Map of East Anglia showing the extent of Chalk aquifers and the superficial till deposits (BGS, 2020) together with catchments in this region that have an NSE score less than the national median of 0.82.**

Autocalibrated-SHETRAN-UK was calibrated using the outlet discharge of the catchment. However, a key objective of this study was to evaluate whether the SHETRAN simulations produce realistic calibrated parameter values, noting that all other parameters in the simulation are pre-set based on measured values from literature. This assessment was conducted by considering simulated transmissivity and conductivity values, both of which exhibited responses consistent with measured data. The evaluation of simulated transmissivity considered the calibrated parameters of deep hydraulic conductivity, shallow hydraulic conductivity, and shallow soil depth. The two additional autocalibrated parameters included the actual/potential evaporation (Ae/Pe) at field capacity, and the urban precipitation fraction. The Ae/Pe ratio accounts for land cover type as well as various groundwater and river abstractions and returns. While some of these abstractions and returns are known and could be incorporated into the model, others remain unregulated and undocumented. Without comprehensive knowledge of all abstractions, assessing the physical realism of this parameter becomes impractical. Future work will investigate the benefits of integrating recently released gridded abstraction data (Rameshwaran et al., 2025) into SHETRAN.



Similarly, evaluating the urban precipitation fraction parameter is not feasible in this context, as it depends on the urban
sewer network, which is not publicly available.

This study utilised daily gridded meteorological driving data and calibrated the model using daily discharge data, which is
sufficient for assessing low flows and comparing results with existing models. However, accurately simulating peak flows,
particularly in smaller catchments, requires sub-daily discharge data. Lewis et al. (2018a) developed gridded hourly rainfall
data for the UK, and quality-controlled hourly discharge data (Fileni et al., 2023) is currently being produced. Future
calibration efforts will integrate this hourly data to enhance model accuracy for peak flow simulations.

Overall, this study demonstrates that SHETRAN, performs comparably to other large-sample national hydrological studies
that employ different model types. The findings suggest that SHETRAN-UK is simulating catchment hydrological processes
well. The model was calibrated using outlet discharge data, and the simulated average catchment transmissivities and
conductivities were then compared to approximate measured values, revealing patterns consistent with observed data. This
strengthens confidence in the model's ability to perform beyond the range of meteorological conditions for which it was
calibrated, which is particularly crucial for predicting future flows under climate change scenarios.

Ongoing work is focused on comparing measured and simulated groundwater levels within SHETRAN-UK catchments
while incorporating the spatial variability discussed above. This effort is being extended to enable the automatic calibration
of the model using both outlet discharge and groundwater levels within the catchment. These enhancements will further
reinforce confidence in the model's ability to produce accurate results for the right reasons.

## 5 Conclusion

The results of this study demonstrate that the autocalibration process significantly enhances the performance of SHETRAN-
UK, a physically-based hydrological model applied at a national scale. The autocalibrated model achieves a higher median
NSE score compared to the uncalibrated version, with substantial improvements observed in catchments where groundwater
processes dominate. Additionally, comparisons with existing national-scale models (Hannaford et al., 2023; Lane et al.,
2019; Lees et al., 2021), indicate that autocalibrated-SHETRAN-UK performs competitively, reinforcing the feasibility of
applying physically-based models on a larger scale when combined with effective calibration strategies.

Unlike conceptual and data-driven models where can be unclear whether the models are producing the right results for the
right reasons, the evaluation of the Autocalibrated-SHETRAN-UK simulated parameters, in this case transmissivity and
conductivity, suggests that Autocalibrated-SHETRAN-UK is achieving accurate results for reasons that mimic the real
world. Consequently, this potentially allows for ungauged catchments to be modelled, by using parameters from



neighbouring catchments. Despite the simplifications required in the model in order to ensure computational efficiency, the autocalibration approach has proven to be effective in optimising key parameters whilst still maintaining hydrological realism. However, a key challenge identified in this study is the difficulty in accurately representing catchments with high spatial variability in soil and aquifer properties. The simplifications used in the autocalibration process leads to limitations in

groundwater-dominated regions, particularly where the Chalk aquifer is overlain by superficial deposits, resulting in poor model performance. However, this study highlights the potential for incorporating spatial variability into future iterations of the model, improving its ability to capture complex subsurface processes.

Future work will focus on refining the model by incorporating groundwater level data into the autocalibration process and

introducing more spatial complexity in soil and aquifer representation within the model to better reflect real-world variability. Overall, this study demonstrates the suitability of autocalibrated, physically-based models for improving national-scale hydrological simulations. Autocalibrated-SHETRAN-UK provides a reliable option, with strong confidence in its ability to perform beyond the range of meteorological conditions used for calibration, highlighting its importance in predicting changes to future flows under climate change scenarios.

**6 Code Availability**

The code supporting the findings of this paper are available from the corresponding author upon request.

**7 Data Availability**

The data supporting the findings of this paper are available from the corresponding author upon request.

**8 Author Contributions**

SJB, EL, ELM & BAS were involved in project conceptualisation and methodology. SJB was responsible for model development, ELM was responsible for running the model simulations and analysing the results, with guidance from SJB and BAS. Data visualisation was undertaken by ELM, with guidance from SJB, EL and BAS. ELM prepared the original paper with contributions from SJB & BAS, and all authors were involved with reviewing and editing of the paper.

**9 Competing Interests**

The authors declare that they have no conflict of interest.



## 10 Acknowledgements

This work is funded as part of the Water Infrastructure and Resilience Centre for Doctoral Training (WIRe CDT) under a grant from the Engineering and Physical Sciences Research Council (EPSRC; grant number EP/S023666/1), in addition to sponsor from Anglian Water and Northumbrian Water Ltd.






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



**Appendix A - 5km grid catchments**

16 catchments greater than 2000km$^2$ are simulated with a grid resolution of 5km. The remaining catchments are simulated with a grid resolution of 1km.

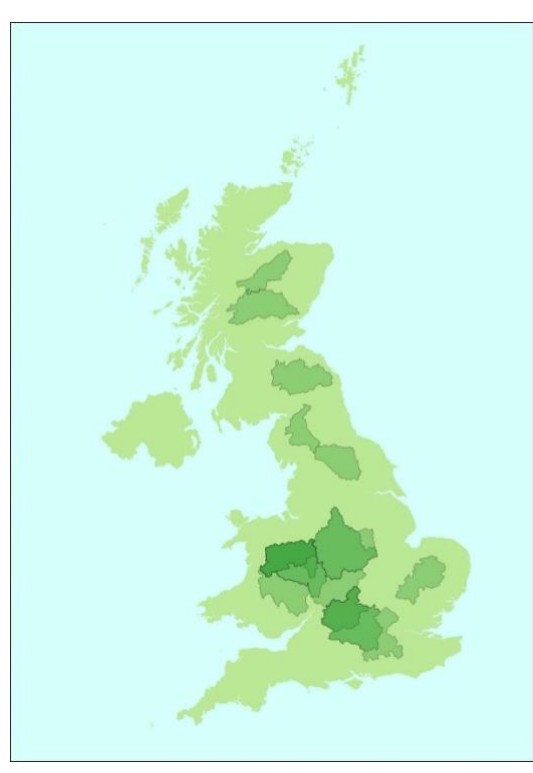

| Catchment | Name | Area (km$^2$) |
|-----------|------|---------------|
| 8006 | Spey at Boat o Brig | 2861 |
| 15006 | Tay at Ballathie | 4587 |
| 21009 | Tweed at Norham | 4390 |
| 27009 | Ouse at Skelton | 3315 |
| 28009 | Trent at Colwick | 7486 |
| 28022 | Trent at North Muskham | 8231 |
| 33035 | Ely Ouse at Denver Complex | 3430 |
| 39001 | Thames at Kingston | 9948 |
| 39002 | Thames at Days Weir | 3444 |
| 39072 | Thames at Royal Windsor Park | 7046 |
| 54001 | Severn at Bewdley | 4325 |
| 54032 | Severn at Saxons Lode | 6850 |
| 54057 | Severn at Haw Bridge | 9895 |
| 54095 | Severn at Buildwas | 3717 |
| 55023 | Wye at Redbrook | 4010 |
| 76007 | Eden at Sheepmount | 2286 |

**Figure A1. Catchment with an area of greater than 2000km$^2$ modelled with a grid resolution of 5km.**




**Appendix B: Uncalibrated-SHETRAN-UK Parameters**

SHETRAN parameters for the uncalibrated model:


Vegetation:

| Vegetation Type | Canopy storage capacity (mm) | Leaf area index | Maximum rooting depth (m) | AE/PE at field capacity | Strickler overland flow coefficient |
|---|---|---|---|---|---|
| Arable | 1 | 0.8 | 0.8 | 0.6 | 0.6 |
| Bare Ground | 0 | 0 | 0.1 | 0.4 | 3 |
| Grass | 1.5 | 1 | 1 | 0.6 | 0.5 |
| Deciduous Forest | 5 | 1 | 1.6 | 1 | 1 |
| Evergreen Forest | 5 | 1 | 2 | 1 | 0.25 |
| Shrub | 1.5 | 1 | 1 | 0.4 | 2 |
| Urban | 0.3 | 0.3 | 0.5 | 0.4 | 5 |

**Table B1. Vegetation parameters for Uncalibrated-SHETRAN-UK.**

Soils:

| Soil Type | Depth at base of layer (m) | Saturated Water Content | Residual Water Content | Saturated Conductivity (m/day) | vanGenuchten-alpha (cm-1) | vanGenuchten-n |
|---|---|---|---|---|---|---|
| Default soil check locally | 1 | 0.403 | 0.025 | 50 | 0.0383 | 1.3774 |
| No Groundwater | 21 | 0.3 | 0.2 | 0.0001 | 0.01 | 5 |
| Peat | 0.3 - 1.2 | 0.766 | 0.01 | 8 | 0.013 | 1.2039 |
| Very Fine | 0.6 - 1 | 0.538 - 0.614 | 0.01 | 8.235 - 15 | 0.0168 - 0.0265 | 1.073 - 1.1033 |
| Fine | 0.3 - 1.2 | 0.481 - 0.52 | 0.01 | 8.5 - 24.8 | 0.0198 - 0.0367 | 1.0861 - 1.1012 |
| Medium Fine | 0.3 - 1.2 | 0.412 - 0.43 | 0.01 | 2.272 - 4 | 0.0082 - 0.0083 | 1.2179 - 1.2539 |
| Medium | 0.3 - 1.2 | 0.329 - 0.439 | 0.01 | 10.755 - 12.061 | 0.0249 - 0.0314 | 1.1689 - 1.1804 |
| Coarse | 0.3 - 1.2 | 0.366 - 0.403 | 0.025 | 60-70 | 0.0383 - 0.043 | 1.3774 - 1.5206 |

**Table B2. Soil parameters for Uncalibrated-SHETRAN-UK.**





Aquifers:

| Aquifer Type | Depth at base of layer (m) | Saturated Water Content | Residual Water Content | Saturated Conductivity (m/day) | vanGenuchten-alpha (cm-1) | vanGenuchten-n |
|---|---|---|---|---|---|---|
| Default low productivity geology check locally | 21 | 0.3 | 0.2 | 0.001 | 0.1 | 5 |
| No groundwater | 20.3 - 41 | 0.3 | 0.2 | 0.0001 | 0.01 | 5 |
| Low productivity aquifer through pores or cracks | 20.4 - 21.2 | 0.3 | 0.2 | 0.001 | 0.01 | 5 |
| Moderately productive aquifer through pores or cracks | 20.3 - 41 | 0.3 | 0.2 | 0.01 | 0.01 | 5 |
| Highly productive aquifer through pores | 20.3 - 41 | 0.3 | 0.2 | 0.1 | 0.01 | 5 |
| Highly productive aquifer through cracks | 20.3 - 41 | 0.3 | 0.2 | 11.5 - 130 | 0.01 | 5 |

**Table B3. Aquifer parameters for Uncalibrated-SHETRAN-UK.**






**Appendix C: Autocalibrated-SHETRAN-UK Parameters**

Bounds for the five auto-calibrated parameters used in Autocalibrated-SHETRAN-UK:

| Parameter | Bounds | Purpose |
|---|---|---|
| Deep soil (aquifer) conductivity | If BFI < 0.8:<br>Initial bounds (m/day): 0.1 - 1<br>Maximum allowed: 0 - 2<br>If BFI > 0.8:<br>Initial bounds (m/day): 0.1 - 100<br>Maximum allowed: 0 - 200 | Controls the rate of water that flows though the deep aquifer. |
| Shallow soil conductivity | Initial bounds (m/day): 1 - 100<br>Maximum allowed: 0 - 200 | Controls the rate of water that flows through the soil layers in a catchment. |
| Shallow soil depth | Initial bounds (m): 0.5 - 4<br>Maximum allowed: 0.1 - 8 | Controls the depth of the soil. |
| AE/PE Ratio – ratio of actual evapotranspiration to potential evapotranspiration in saturated soils | Initial bounds (-): 0.5 - 2<br>Maximum allowed: 0.1 - 4 | Accounts for the actual evaporation varying according to the vegetation type. In addition, it is used to correct errors in the overall water balance, for example when groundwater abstractions are present, or the surface water and groundwater catchment extents do not match. |
| Urban precipitation fraction | Initial bounds (-) 0.1 - 0.5<br>Maximum allowed: 0 - 1 | Splits urban precipitation into two fractions as a percentage; one where precipitation flows directly into stormwater drains, and one where precipitation is removed due to flowing into combined sewers. This based on some of the techniques developed in (Birkinshaw et al., 2021). |

**Table C1. Bounds for 5 autocalibrated parameters in Autocalibrated-SHETRAN-UK.**







Non-calibrated parameters in Autocalibrated-SHETRAN-UK:

| Parameter Type | Parameter | Value |
|---|---|---|
| Soil | Effective Porosity (Saturated Water Content – Residual Moisture Content) | 0.42 (-) |
| | Van Genuchten alpha parameter | 0.0083 (cm-1) |
| | Van Genuchten n parameter | 1.25 (-) |
| Aquifer | Effective Porosity (Saturated Water Content – Residual Moisture Content) | 0.10 (-) |
| | Van Genuchten alpha parameter | 0.01 $(cm^{-1})$ |
| | Van Genuchten n parameter | 5.0 (-) |
| Rural vegetation | Canopy Storage capacity | 3.0 (mm) |
| | Canopy Drainage Ck parameter | $1.4 \times 10^{-5}$ (mm s$^{-1}$) |
| | Canopy Drainage Cb parameter | 5.1 (mm$^{-1}$) |
| | Rooting Depth | 1.0 (m) |
| Overland flow | Stricker Coefficient for rural cells | 2.0 ($m^{1/3}$/s) |
| | Stricker Coefficient for urban cells | 12.0 ($m^{1/3}$/s) |
| | Stricker Coefficient for rivers | 20.0 for catchments < 1000km$^2$ ($m^{1/3}$/s) 50.0 for catchments > 1000km$^2$ ($m^{1/3}$/s) |
| Snow | Degree day Factor | $2 \times 10^{-4}$ (mm/s/°C) |
| River channel generation | Number of upstream grid squares needed to produce a river channel | 2 (-) |
| | Drop from grid square elevation to channel bed elevation | 2 (m) |
| | Minimum elevation drop between connecting channels | 0.5 (m) |

**Table C2. Non-calibrated parameters in Autocalibrated-SHETRAN-UK.**






**Appendix D: Details of the automatic calibration**

The autocalibration process of SHETRAN is based on the Shuffled Complex Evolution method, developed at the University

of Arizona as outlined in (Duan et al., 1992, 1994, 1993). The SCE-UA method begins with the random sampling of a

population of points from within the feasible parameter space. This provides the potential for locating the global optimum

without being biased by pre-specified starting points (Duan et al., 1994). This randomly sampled population is divided into

several 'complexes', with each complex containing $2n+1$ points, where $n$ is the number of parameters being used in the

calibration process. The partitioning of the population into complexes allows a freer and more extensive exploration of the

feasible parameter space in different directions, therefore acknowledging that there is a possibility that the problem has more

than one region of attraction. Each of these complexes 'evolves' based on a statistical reproduction process that uses the

shape of the simplex to direct the search in an appropriate direction. Throughout the process at specified intervals, the entire

population is reshuffled, with points being reassigned to complexes. This enhances the survivability by ensuring information

about the search space gained independently by each complex is shared (Duan *et al.,* 1994).


As this process is iterative, and the point in parameter space where the optimum value of the objective function is achieved is

unknown and may never be found, the algorithm for calibration requires some sort of stopping criterion. The most

commonly used within literature have been: objective function convergence criterion, parameter convergence, and maximum

iteration number (Zhang et al., 2015). Parameter convergence criterion is the most suitable for enabling parameter

optimisation as the algorithm terminates when the parameters are not obviously changing, however in order to avoid wasting

computational time, maximum iteration criterion was used within this study. Consequently, based on the SCE-UA method

and the maximum iteration criterion, the number of model runs within the autocalibration process is equal to 462. The

calculation of this is outlined below.

$\boldsymbol{NoP} = Number\ of\ complexes = \boldsymbol{2}$

$\boldsymbol{NoN} = Number\ of\ points\ (parameters)\ in\ each\ complex = \boldsymbol{5}$

$\boldsymbol{NoIter} = Number\ of\ iterations\ of\ algorithm = \boldsymbol{20}$

$\boldsymbol{NoM} = Number\ of\ members\ in\ each\ complex = 2 \times NoN + 1 = \boldsymbol{11}$

$\boldsymbol{NoBeta} = Number\ of\ steps\ taken\ by\ each\ complex\ before\ being\ reshuffled = 2 \times NoN + 1 = \boldsymbol{11}$

$\boldsymbol{NoS} = Number\ of\ initial\ sample\ runs = NoP \times NoM = \boldsymbol{22}$

$\boldsymbol{Number\ of\ model\ runs\ in\ autocalibration} = NoS + \big((NoP \times NoBeta) \times NoIter\big) = 22 + \big((2 \times 11) \times 20\big)$

$= \boldsymbol{462}$






To limit the amount of time taken to carry out the autocalibration process, 121 runs were carried out for the 16 largest catchments (Appendix A), with NoP = 1 and NoIter=10.