# Peer review of "Autocalibration of a physically-based hydrological model: does it produce physically realistic parameters?"

_EGUsphere, 2025_

## Author Comment (AC1)

**Reply to Reviewers' Comments**

**Key:**

Reviewer comment.

Response.

**Reply to Reviewer #1**

**General comments:**

The paper presents the results of a calibrated process-based model and compares them with other models. Moreover, the paper describes qualitatively the similarity of the hydrogeological parameters versus other sources of information. The improvement is substantial compared with the uncalibrated model, which sets the calibrated model at the level of conceptual and machine learning models, with the addition of having the interpretability of its parameters.

We thank the Reviewer for their constructive and encouraging summary of our work. We appreciate the time and effort they have invested in evaluating manuscript and providing feedback that will certainly help to improve the clarity and presentation of the results.

Two main concerns emerge from the paper. First, it is well known by the hydrology community that uncalibrated models can improve substantially if some calibration of their parameters is applied. Therefore, the author should put more effort into highlighting the difficulty of applying a calibration in such models. In this context, a better description of the process applied during the calibration will benefit a broader community that needs to calibrate those models. However, when the methodology was finally mentioning the autocalibration, the authors sent the information to the appendix, which goes in detriment of the importance of presenting a methodology for calibrating such models. From my point of view, this is a key point that is not presented adequately.

We thank the Reviewer for articulating this important point. We agree that the challenges associated with calibrating physically-based models, as well as the specific methodology applied in our autocalibration approach, deserves greater emphasis in the main manuscript. In response, we will move the autocalibration description from the appendix into the main body of the paper and will expand it to provide a more detailed explanation of the steps involved, the computational challenges, and the rationale for the chosen approach. We believe this revision will make the methodology more accessible and useful to the broader community.

The second main concern is about the "realism" of the parameters. The authors spent many sections of the paper trying to probe that, but they did so qualitatively. Proving this strong argument needs more than comparing maps. The authors can analyze the results with scatter

plots, correlations, % of catchments with consistent parameters, developing parameter maps with Krigging (or cokriging), etc., just to mention more robust analyses. Without that analysis is impossible to answer the title question. Moreover, the authors mention the word "real" many times to refer to the comparison with other maps. However, they fail to understand that such maps are just the result of some model. Therefore, they cannot be considered as "real" data. If they want to compare with real data, they should compare with the parameters extracted from the wells, therefore, point observations. Any other spatial distribution of such parameters is just a model (synthetic data). Another situation that the author did not mention about the groundwater parameters is that they are probably the parameter with the most uncertainty. The authors should incorporate an analysis of other parameters that are probably more easily constrained by observation or using remote sensing products.

We thank the Reviewer for raising this important concern regarding the assessment of parameter "realism". We accept the view that qualitative map-to-map comparisons alone are insufficient to fully evaluate the plausibility of the calibrated parameters. In response, we will substantially expand our analysis.

Firstly, we will add groundwater level evaluation into analysis, comparing simulated groundwater-levels with groundwater level observations. As suggested by the Reviewer, this will strengthen the assessment of model behaviour as it is a test against "real" observed groundwater behaviour. This will make use of the recently released CAMELS-GB v2 (Coxon et al. 2025) dataset which contains groundwater level time series for 55 groundwater wells.

Secondly, for deep soil conductivity, we will translate available observational datasets into average catchment scale values and include a scatter plot comparison with the simulated values in the revised manuscript. While we acknowledge that some national datasets originate from models rather than direct measurements, these remain the only spatially comprehensive sources available. Therefore, supplementing these dataset comparisons with analyses based on measured groundwater levels in selected catchments provides an independent line of evidence.

We believe these additions will provide a more robust and balanced assessment of parameter plausibility and will strengthen our ability to answer the research question posed in the title.

In summary, I consider that the paper has results that are valuable for the hydrology community, but major changes must be made to highlight the points that are important for such a community, moving away from just presenting the result of a model.

We thank the Reviewer for their overall assessment and for highlighting the need to better emphasise the broader hydrological relevance of our findings. In response, we will expand the discussion to more clearly articulate the implications of the results for the hydrology community, going beyond the presentation of model outputs. This will include a stronger focus on what the calibrated model reveals about catchment behaviour, parameter plausibility, and the challenges and opportunities associated with calibrating physically based models at national scale. We believe these additions will improve the manuscript's clarity and relevance.

**Minor comments:**

Line 11-12.      That is a very strong argument that will create a lot of controversy. I like process-based models because they can represent a very complicated world with their simplified equations. However, they do not represent the truth because the complexity of a catchment is many orders of magnitude more than the representation of a process-based model. Moreover, in general, given that machine learning models can generate better results than process-based models under the same data, it shows us that there is more to improve in such models. Therefore, I don't think the authors need to enter into this controversy.

Thank you for highlighting this. We agree that the original wording may imply a stronger position than intended and could be interpreted as entering an unnecessary controversy. We will therefore modify the text to reflect this comment in the revised manuscript.

Line 27-29.      I do not think talking about future work in the abstract is a good idea because authors should summarize their findings, not what they did not do.

We thank the Reviewer for this remark. We agree and will remove this in the revised manuscript.

Line 50-52.      This statement could be easily said too for a process-based model. To generate the equations used in the model, a lot of data and years of experiments were needed; therefore, saying that data-based models require more data is not fair.

We thank the Reviewer for highlighting this point. We will think about revising the text to more accurately reflect this perspective in the revised manuscript.

Line 56.              "well-established physical laws". I am pretty sure that the only physical laws implemented in the model are mass and energy conservation. Moreover, these physical laws are probably not satisfied at the resolution of the model. Any other equations used by the model are just simplifications of the truth.

We thank the Reviewer for this remark. We will amend this to be more precise.

Line 59.              The degree of uncertainty of such parameters is huge when they are used in catchment-scale models (or at 1km$^2$ resolution). This statement does not have support.

We thank the Reviewer for this comment and appreciate the opportunity to clarify our intent. We agree that parameter uncertainty is substantial when applied at catchment scale, but note that the statement in Line 59 is made within the general context of physically based modelling, rather than specifically referring to how SHETRAN is applied in this study. For this reason, we believe the statement is appropriate in the introduction; however, we recognise that it would benefit from clearer support. We will also expand the discussion section to further highlight this issue.

Line 125.         The idea of "realism" is oversold. It is well known that parameters do not necessarily represent something real in the world, especially if we work at 1km$^2$ resolution.

We thank the Reviewer for this helpful comment. We fully agree that, particularly at a 1 km$^2$ resolution, model parameters should not be interpreted as literal representations of real-world physical properties and acknowledge the long-standing discussion on the uncertainty of parameters when used at a coarse scale in physically-based hydrological models (e.g., Beven, 1989). Our intention was not to imply otherwise, but rather to highlight that national-scale studies in the UK rarely examine the parameter values produced by simulations, focusing instead almost exclusively on streamflow performance. We will revise the manuscript to clarify this point and to avoid suggesting that parameter realism is expected, only that its evaluation is generally overlooked.

> "In hydrological simulations using the CAMELS large-sample dataset, model evaluation has typically focused on reproducing outlet discharge, with limited attention to the behaviour or plausibility of the resulting parameter values, regardless of model resolution."

Line 143.         It would be beneficial if more details about the variability were added. The variability in the CAMELS dataset of GB, US, and CL is very different.

We thank the Reviewer for this valuable suggestion. We agree that the variability across the CAMELS-GB, CAMELS-US, and CAMELS-CL datasets differs substantially. However, because our study does not directly compare these datasets or rely on their differences for the core analysis, we feel that providing additional detail on this variability would not substantially strengthen the manuscript. We will, however, ensure that the text clearly acknowledges that these datasets differ in their hydrological and climatic characteristics.

Line 181.         Does it mean you treated the catchment as lumped? Is the meteorological forcing lumped too?

We thank the Reviewer for this question. In our implementation within the autocalibration process, the model parameters are treated in a 'lumped' manner; however, the meteorological data is not lumped and is applied with spatial variability. We will revise the manuscript to clarify this distinction.

Line 205.         The first word in the title is about the autocalibration, so the reader will think that something novel is presented about that in the paper. However, the authors did not talk about that and sent it to the appendix. This method should be highlighted more.

We thank the Reviewer for this helpful observation. We agree that, given the prominence of autocalibration in the title, the method should be described more clearly in the main text. We will therefore move the autocalibration method description from the appendix into the main manuscript and expanded the explanation to highlight its role in the study.

Line 281.          The paper must be general enough for a worldwide audience; therefore, referencing places must be avoided.

We thank the Reviewer for this comment. While we agree that the paper should be accessible to a worldwide audience, we believe that referring to specific regions within the UK is important for scientific clarity. The spatial patterns we describe, such as differences in performance in Chalk-dominated catchments or regions near the Welsh border, are directly tied to underlying hydrogeological characteristics. These locations are widely recognised in hydrological literature and are essential for interpreting why model behaviour varies across catchments.

Line 282-286.   A section cannot be referenced before it is introduced in the text.

We thank the Reviewer for noting this. We will remove the premature reference and relocate the relevant text to the appropriate section where it is first introduced.

Line 295.          Where are the PBIAS results presented in the paper?

We thank the Reviewer for pointing this out. We will include the PBIAS results in the appendix and reference them appropriately in the main text.

Line 297.          Add reference to the BGS hydrogeological aquifer map.

We thank the Reviewer for this suggestion. We will add the appropriate reference to the BGS hydrogeological aquifer map in the revised manuscript.

Figure 4.          This information is better represented by a scatter plot.

We thank the Reviewer for this suggestion and will consider the use of a scatter plot; however, we feel that such a representation may not fully support the spatial focus of our analysis across the UK, particularly given the small number of catchments underlain by highly productive aquifers that perform to the standard being looked at in this Figure. For this reason, we believe the current spatial representation remains informative. Nevertheless, we will review the figure and evaluate whether an additional scatter-plot representation could complement the existing analysis.

Line 319-322.   This statement is not supported by the results, or at least by the presented figures. Add some reference, or you must clarify that this is just a hypothesis.

We thank the Reviewer for this observation. We will add an appropriate reference to support this statement in the revised manuscript.

Figure 5.          The same color must be used for positive (reds) and negative (blues) changes. Blue cannot be used for positive changes.

We thank the Reviewer for this comment. We will revise Figure 5 to ensure consistent colour conventions by removing the use of blue for positive values.

Line 335.          Why are the CAMELS attributes not used?

The catchment descriptors used here are equivalent to the corresponding CAMELS attributes, and the results do not depend on which is used. We will modify the manuscript to highlight that that similar CAMELS attributes are available.

Line 344.          There are a few catchments with NSE lower than 0.6; therefore, there is not enough information to support this statement.

We thank the Reviewer for this comment. We acknowledge that a small number of catchments have NSE values lower than 0.6. To ensure that our statement is properly supported, we will revise Figure A by removing the log-transformed x-axis (as shown below). We will clarify this point in the manuscript to ensure the interpretation is properly supported.

[Figure]

Line 358.          This is good, but a figure is not needed to show that no relationship was found. Send the figure to the appendix.

We thank the Reviewer for this suggestion. We will move the figure to the appendix as recommended.

Line 360.          I think this analysis would be more relevant if the best ML and lumped model were included.

We thank the Reviewer for this helpful suggestion. We will consider incorporating the best-performing machine learning and conceptual models into this analysis to enhance the

relevance and comparability of the results. We will add these comparisons in the revised manuscript.

Line 378-380.   Is this statement checked by changes in the model (sensitivity analysis), or is it just a statement assumed, given the simplifications of the calibration?

We thank the Reviewer for this comment. This statement is based on assumptions arising from the simplifications made during the calibration process rather than from a dedicated sensitivity analysis. We will revise the manuscript to make this clear.

Line 317.          Why simpler? The calibration method used is equally simple to many of the models presented.

We thank the Reviewer for this comment. We are unsure which part of the text is being referred to here, as Line 317 does not mention anything being "simpler." We would be grateful for clarification so that we can address the concern appropriately. If the intended reference is Line 417, we will remove the word "simpler" from the subheading in the revised manuscript.

Figure 8.          This figure does not have a fair comparison between models because each study has different catchments (probably different training periods). However, only one autocalibrated result is presented. The result of the autocalibrated model for each study must be added.

We thank the Reviewer for this comment. The Reviewer is correct that the studies use different sample sizes and calibration/validation periods, which makes direct comparison challenging. As noted in the figure caption, these concerns are highlighted and only catchments common to all studies were included in the comparison. Nevertheless, we recognise that the current presentation may still lead to misinterpretation. We will therefore revise Figure 8 to clarify the basis of the comparison and improve its presentation so that the limitations are more explicit. The revised manuscript will reflect these changes accordingly.

Line 442.          Why were these parameters selected? They are probably the most uncertain of all of them.

We thank the Reviewer for this comment. These parameters were selected because they relate directly to the subsurface processes, which were the focus of the autocalibration procedure. Although we acknowledge that subsurface parameters can be among the most uncertain, they are also the parameters for which calibration was specifically applied in this study. We will clarify this rationale in the revised manuscript.

Line 443.          The main agreement is in the southern areas. The rest of the catchments are not necessary in agreement. Try to qualify the number of catchments in agreement or disagreement.

We thank the Reviewer for this comment. We agree that the strongest agreement between the autocalibrated transmissivity values and the BGS aquifer designations occurs in the southern regions of the UK, particularly across the major Chalk aquifers. Outside of these areas, the agreement is more variable. In response, we will attempt to quantify the number of catchments showing agreement and disagreement across all regions to provide a clearer and more objective assessment. This will be added to the revised manuscript to ensure the statement is fully supported by the results.

Figure 10.        Color over color is not the best way to present the information. Try to incorporate a hatch for the catchments.

We thank the Reviewer for this helpful suggestion. We will revise Figure 10 to incorporate hatching or another alternative for the catchments so that the information is clearer and avoids colour-on-colour overlap.

Figure 12.        A comparison only with maps is not enough to show the similarities between the parameters and the external source. A scatter plot would be a better option. Try to incorporate the NSE as color to have another dimension for the differences.

We thank the Reviewer for this suggestion. We will revise Figure 12 by presenting the comparison as a scatter plot and incorporate NSE as a colour dimension to provide additional insight into the relationship between the parameters and the external data source.

Line 529.        The term autocalibrated was used without describing it. What makes the model autocalibrated or just calibrated?

We thank the Reviewer for this comment. In the manuscript, the term *autocalibrated* refers specifically to the calibration procedure developed and applied in this study, as introduced earlier in the introduction. To avoid ambiguity, we will revise the text to make this explicit at the first use in the discussion section.

Line 536-537.   I disagree. One analysis of performance and characteristics was presented, which was not conclusive. Later, only groundwater parameter analysis is presented without applying a direct comparison with performance.

We thank the Reviewer for this observation. We acknowledge that the initial analysis linking model performance to parameter characteristics was limited in scope and not fully conclusive. In response, we will expand this analysis to provide a clearer and more direct examination of the relationship between performance and the calibrated groundwater parameters. This will include incorporating scatter plots and additional quantitative comparisons, as suggested and described earlier, to strengthen the connection between parameter behaviour and model performance. We will revise the manuscript accordingly to ensure that the conclusions are properly supported by the presented results.

Line 544-544.   Where did you present these results?

We thank the Reviewer for this comment. These results will be explicitly added to the results section in the revised manuscript to ensure they are clearly presented and properly referred to.

Line 546-548.   The authors are leaking information about some results they did not present in the paper. It is nice that more results would be available, so the authors know the reason, but the discussion and conclusion must be associated only with the results presented in the paper.

We thank the Reviewer for this comment. We appreciate the principle that discussion and conclusions should be grounded in the results presented. At the same time, we believe that a discussion section also has an important role in situating the study within the broader field, acknowledging limitations, and outlining implications for future work. These elements often extend beyond the specific figures and tables included, while still being firmly informed by the study's findings.

To address the Reviewer's concern, we will consider revising the text to ensure that all interpretative statements remain clearly connected to the presented results, while retaining the broader reflective components that help clarify the study's contribution, limitations, and potential avenues for further research.

Line 566.          "measured data". That is not true. The authors compared with external sources that used observations to create a spatial distribution of the parameters. However, this is far from observed data. The author can calculate the density of observations used in such maps, and probably they will be less than one per catchment. Therefore, the map is just an interpolation method with high uncertainty that cannot be considered as "measured data".

We thank the Reviewer for this important clarification. We agree that national-scale parameter maps derived from modelling or interpolation cannot be considered "measured data" in a strict sense, particularly given the low density of underlying observations, however these remain the only spatially comprehensive sources available. In the revision, we will avoid referring to these datasets as measured data and instead describe them explicitly as products derived from point observations and subsequent spatial interpolation or modelling.

Where transmissivity is concerned, we will highlight that estimates originate from point well observations. As noted earlier, we will also supplement these comparisons with analyses based on observed groundwater levels in selected catchments, providing an additional and independent line of evidence.

Line 569.          Several remote sensing products could be used to analyze Ae/Pe in the model, but they were not used in the analysis.

We thank the Reviewer for this suggestion. We agree that incorporating remote sensing products to analyse Ae/Pe would provide valuable additional insight. However, integrating these datasets is beyond the scope of the current study, which is focused primarily on evaluating the effects of autocalibration on model performance and parameter behaviour. We will note this as a potential avenue for future work in the revised manuscript.

Line 578-579.   The statement about the sub-daily scale is true; however, GR4J and LSTM models were able to predict better than SHETRAN using lumped data. Therefore, the sub-daily scale will not solve the problems in the architecture that SHETRAN could have.

We thank the Reviewer for this insightful comment. We agree that moving to sub-daily simulations will not, on its own, resolve structural limitations in SHETRAN or guarantee improved performance relative to models such as GR4J or LSTM, which have demonstrated strong predictive skill even with lumped inputs. Our intention was not to imply that finer temporal resolution would overcome architectural challenges, but rather that SHETRAN is designed to represent processes operating at sub-daily timescales, and that evaluating the model at such resolutions may provide a clearer indication of its strengths and weaknesses, particularly for smaller catchments, where water moves through the system more quickly and sub-daily dynamics become more important.

We will revise the text to clarify this distinction and avoid overstating the potential benefits of sub-daily simulations.

Line 586.          "approximate measured values", "observed data". This is not true. See comment in line 566.

We thank the Reviewer for pointing this out. In line with our response to Comment 566, we will amend the revised manuscript to avoid terms such as "approximate measured values" and "observed data" when referring to spatially interpolated or model-derived products. Instead in the revised manuscript, we will describe these explicitly as datasets derived from limited point observations and subsequent interpolation or modelling, and we will more clearly acknowledge their associated uncertainties.

Line 587-588.   The author did not test for extended future scenarios or climate change; therefore, they cannot be confident about that.

We thank the Reviewer for this comment. We agree that, since future climate change conditions were not directly tested in this study, we should avoid implying validated performance under such conditions. Our intention was instead to convey that the model's ability to perform well across a wide range of historical meteorological conditions provides *greater confidence* that it may behave robustly when applied to non-stationary future climates, whilst recognising that this remains a hypothesis rather than a demonstrated result.

We will revise the text to clarify that this confidence refers to the model's demonstrated skill under diverse historical conditions, and not to tested performance under future climate scenarios.

Line 590-593.   This is information that is not relevant to this paper. The authors should focus only on the results presented.

We thank the Reviewer for this comment. We will remove this material from the revised manuscript.

Line 605. "real world". The authors are overselling their result. They just compared the result visually with maps generated by other sources. This is not enough to probe consistency. For example, they did not check for spatial consistency between catchments. How can the "real world" be mentioned if the parameters change drastically between adjacent catchments?

We thank the Reviewer for this important comment. We agree that referring to the "real world" is too strong given that our comparisons were made primarily against externally generated maps rather than direct observations, and that spatial inconsistencies between neighbouring catchments highlight the limits of such an assessment.

We will revise the wording to avoid the phrase "real world" in the amended manuscript. In addition, as noted earlier, we will strengthen the analysis by incorporating scatter plots, quantitative comparisons, and checks for spatial consistency between catchments to better support any statements about parameter plausibility.

**References:**

Coxon, G., Zheng, Y., Barbedo, R., Cooper, H., Fileni, F., Fowler, H. J., Fry, M., Green, A., Gribbin, T., Harfoot, H., Lewis, E., Neto, G. G. R., Qiu, X., Salwey, S., and Wendt, D. E.: CAMELS-GB v2: hydrometeorological time series and landscape attributes for 671 catchments in Great Britain, Earth Syst. Sci. Data Discuss. [preprint], https://doi.org/10.5194/essd-2025-608, in review, 2025."

Beven, K. (1989). Changing ideas in hydrology—the case of physically-based models. Journal of hydrology, 105(1-2), 157-172.

---

## Author Comment (AC2)

**Reply to Reviewers' Comments**

**Key:**

Reviewer comment.

Response.

**Reply to Reviewer #2**

General Comments:

This work uses the SCEUA automatic calibration platform to calibrate surface water flows predicted over a number of watersheds in the UK. The SHETRAN distributed model is used as the simulation platform, run daily and compared to daily streamflow values. While I appreciate the goal of arriving at the right answers for the right reasons, the analysis to establish the physical basis for the calibrated values is insufficient to establish this. Furthermore, the approach used to conceptualize each model is very similar if not identical to the watershed models used to contrast the work done here. I think more work is needed along with revisions to the manuscript to establish these points. Comments are below.

We gratefully thank the Reviewer for the evaluation of our work and also for the constructive comments, the corrections and suggestions provided will certainly help to improve the manuscript to make it ready for publication. These are discussed below.

* what are the limitations and biases in the shallow subsurface? The authors state that a single aquifer unit 20m in thickness was used; this is only shallow surficial, unconfined groundwater. The fixed 20m thickness has an influence on the transmissivity estimates. The authors should explore this sensivity with some number of representative watersheds to ensure this does not impart bias.

We thank the Reviewer for this valuable comment and suggestion. We agree that the fixed 20 m aquifer thickness represents a simplification of the shallow subsurface and may influence the transmissivity estimates. In response, we will explore this sensitivity further by running a set of representative catchments with an increased aquifer thickness of 50 m. We will assess the impact on model performance (e.g., NSE) as well as on the resulting transmissivity values. The outcomes of this sensitivity analysis will be included in the revised manuscript to evaluate any potential bias introduced by the fixed aquifer thickness.

*The authors state that some watersheds were modeled with 1km columns while others were modeled with 5km columns. This is a substantial and seemingly arbitrary adjustment in resolution. What are the limitations of the grid resolution switch for the 16 catchments run at 5km? Did the authors conduct a sensitivity study on resolution to establish these final

values? Given that topography will be smoothed out substantially at 5km, a sensitivity study to ensure that results might be transferrable between these different resolutions is needed.

We thank the Reviewer for this comment. Sixteen catchments were run at a 5 km grid resolution because they were too large to be simulated feasibly at 1 km within the autocalibration framework. For example, the 54057 catchment (Severn at Haw Bridge) has a catchment area of 9895 km$^2$ and a single simulation at 1km resolution took approximately 2 days to complete. Consequently, using a coarser 5 km grid was the only practical option for including these catchments in the national-scale analysis.

We acknowledge that moving to 5 km resolution smooths topography and subsurface structure and may affect hydrological processes, and we agree that this introduces limitations. While a full resolution-sensitivity analysis across all large catchments was not feasible due to computational constraints, we will add discussion to the revised manuscript outlining these limitations explicitly and explaining how grid resolution may influence the transferability and comparability of results between the 1 km and 5 km simulations. We will also include a summary of simulation times to clarify the computational constraints underpinning this choice.

*It is well known the groundwater does not follow surface topographic divides and aquifer systems connect watersheds laterally. It appears that each of these watersheds is modeled independently which does not allow for groundwater import and export between watersheds along with regional groundwater flow. Can the authors show that lateral groundwater flow does not impact their results?

We thank the Reviewer for raising this important point. We recognise that groundwater systems do not always follow surface topographic divides and that lateral groundwater flow between neighbouring catchments can occur. In this study, each of the 698 catchments was modelled independently; however, the catchment boundaries used are those defined by the National River Flow Archive (NRFA), which has undertaken detailed assessments to align surface-water and groundwater boundaries wherever possible. According to the NRFA documentation, only 7 of the 698 catchments have known mismatches between surface and groundwater divides.

We agree that lateral groundwater exchanges may occur in some locations, but given the national scope and the independence of the model runs, it is not possible to explicitly simulate cross-catchment groundwater flow or quantify its effect within this framework. To address this limitation transparently, we will include in the appendix a list of the catchments where the NRFA identifies discrepancies between surface and groundwater boundaries and will discuss the implications of these limitations in the revised manuscript.

*The authors visually compare the transmissivity values produced at the end of the calibration process with BGS aquifers (Fig 11 and 12). Visually, the maps appear to bear no similarity. The other continental scale efforts the authors mention (e.g. Naz et al, Yang et al) start with geologic maps and aquifers to parameterize their model, then adjust parameter values accordingly. 1. Why was this more common approach not taken? 2. What quantitative steps can be taken in the manuscript to show any agreement between the geologically derived transmissivity values and the ones arrived at from this current study?

We thank the Reviewer for this detailed and constructive comment.

Regarding Figures 11 and 12, we believe that the transmissivity comparison in Figure 11 provides significant value to the paper and discussion of the parameters produced. Figure 12, however, presents deep-soil conductivity, not transmissivity, and we agree that both figures can be improved to make the comparisons clearer. We will therefore remake and revise these figures in the updated manuscript to enhance their interpretability and ensure that the analysis is presented more transparently.

1. Why the more common geology-based parameterisation approach was not taken

The Reviewer is correct that many continental-scale modelling studies (e.g., Naz et al., Yang et al.) initialise groundwater parameters directly from geological or aquifer maps before calibration. This *was* the approach taken in Uncalibrated-SHETRAN-UK, which uses hydraulic conductivity values from the *Aquifer Properties Manual* (Allen et al., 1997; MacDonald and Allen, 2001, Lewis, 2016). Autocalibrated-SHETRAN-UK, in contrast, departs from this by calibrating subsurface parameters directly, allowing the model to adjust parameters based on hydrological behaviour rather than relying solely on the mapped geology.

We will make this distinction more explicit in the revised manuscript to avoid confusion.

2. Quantitative steps to show agreement with geologically derived transmissivity

Figure 11 compares transmissivity estimates from point well observations (Allen et al., 1997) with the transmissivity values obtained from this study. We acknowledge that this was not sufficiently clear, and we will improve the figure and text to ensure that the comparison is transparent and easily interpretable.

In addition, to strengthen this part of the analysis we will incorporate:

- scatter-plot comparisons between calibrated transmissivity and observational estimates,

- NSE as a colour dimension, and

- Groundwater level observation comparisons

Together, these revisions will provide a more robust and quantitative assessment of parameter plausibility.

*It would appear that a major component of the physics based approach would be to generate water table depth values and ET estimates along with streamflow. These values could help characterize change in storage and ET fluxes, which along with streamflow discharge would close the water budget for each watershed potentially limiting the space of equifinality. Given the unconstrained nature of the calibration exercise and simple model configuration, it appears that many solutions might provide the same streamflow estimates. A very thick soil layer could be produced that mimics the aquifer or AE/PE values could change the water budget. These could easily take on unrealistic bounds, detracting from the central theme of the work. The authors should justify the use of just streamflow in calibration. More discussion is needed, perhaps also with sensitivity cases, to ensure that parameter values are not unrealistic.

We thank the Reviewer for this comment. We used streamflow-only calibration to remain consistent with other national-scale studies (e.g., Lane et al., 2019; Lees et al., 2021), none of which incorporate groundwater or ET observations in their calibration procedures. We will clarify this in the revised manuscript.

We agree that relying solely on discharge increases the potential for equifinality. To limit this, only five subsurface parameters were calibrated, each with strict, limiting and realistic bounds (see Appendix C). In addition, the development of the parameters is not entirely unconstrained during the autocalibration as the mass balance is conserved. In the revised manuscript, we will expand the discussion on these limitations and include additional analyses (e.g., comparisons with groundwater levels and transmissivity estimates) to help demonstrate that the resulting parameter values remain plausible.

*There are no surface water parameters, such as channel width or mannings roughness coefficients included in the analysis or sensitivity. Is this because channel routing was somehow not included in the study or for some other reason?

We thank the Reviewer for this comment and question. Non-calibrated parameters relating to surface water flow, such as the Strickler Coefficent for both overland flow and channel routing are presented in Appendix C. Channel routing is included as the model using diffusive approximation of the St Venant equations (Ewen et al., 2000).

**References**

Ewen, J., Parkin, G. and O'Connell, P.E. (2000). SHETRAN: Distributed River Basin Flow and Transport Modelling System. ASCE J. Hydrologic Eng., 5, 250-258.